# SALSA VERDE: a machine learning attack on Learning With Errors with sparse small secrets

**Cathy Yuanchen Li**
FAIR, Meta

**Emily Wenger**
The University of Chicago

**Zeyuan Allen-Zhu**
FAIR, Meta

**Francois Charton**[*]
FAIR, Meta

**Kristin Lauter**[*]
FAIR, Meta

## Abstract

Learning with Errors (LWE) is a hard math problem used in post-quantum cryptography. Homomorphic Encryption (HE) schemes rely on the hardness of the LWE problem for their security, and two LWE-based cryptosystems were recently standardized by NIST for digital signatures and key exchange (KEM). Thus, it is critical to continue assessing the security of LWE and specific parameter choices. For example, HE uses secrets with small entries, and the HE community has considered standardizing small sparse secrets to improve efficiency and functionality. However, prior work, SALSA and PICANTE, showed that ML attacks can recover sparse binary secrets. Building on these, we propose VERDE, an improved ML attack that can recover sparse binary, ternary, and narrow Gaussian secrets. Using improved preprocessing and secret recovery techniques, VERDE can attack LWE with larger dimensions ($n = 512$) and smaller moduli ($\log_2 q = 12$ for $n = 256$), using less time and power. We propose novel architectures for scaling. Finally, we develop a theory that explains the success of ML LWE attacks.

## 1   Introduction

Language models have been successfully applied to numerous practical science problems in recent years. For example, transformers [57] have been used to solve problems in mathematics [40, 24], theoretical physics [26], chemistry [50], and biology [46]. In this paper, we present an application of transformers to computer security: the cryptanalysis of Learning With Errors (LWE) [45], a hard math problem underpinning leading proposals for post-quantum public key cryptography.

**Public-key cryptosystems** are the main solution for secure communication over the Internet. Public keys can be used to encode messages or verify digital signatures to or from a user with the corresponding private key. Security relies on the fact that recovering the private key from the public data requires solving a computationally hard math problem. Most currently deployed public-key systems are based on RSA [47], which relies on the hardness of factoring large numbers into products of primes. Unfortunately, large-scale quantum computers will enable implementation of Shor's algorithm [52], which can factor integers in quantum polynomial time and break such systems. As a result, new hard problems are sought to serve as the basis of post-quantum public key cryptography (PQC). The US National Institute of Standards and Technology (NIST) ran a 5 year competition to define future PQC standards [2], and standardized 4 PQC systems in July 2022 [17]. Two of these rely on special cases of the same hard (and Shor-free) problem: Learning with Errors (LWE).

**The Learning With Errors** problem (LWE) assumes that it is hard to recover a secret vector $\mathbf{s}$, given many LWE samples $(\mathbf{a}, b)$. In a LWE sample, each $\mathbf{a}$ is a random vector of $n$ integers modulo $q$ ($n$ is the dimension and $q$ is the modulus), and $b$ is a noisy modular inner product of $\mathbf{a}$ and the secret key $\mathbf{s}$—that is, $b = \mathbf{a} \cdot \mathbf{s} + e \mod q$, with the error $e$ drawn from a Gaussian distribution of

---

[*]Co-senior authors

width $\sigma_e$ centered at 0, $e \sim N(0, \sigma_e^2)$ and $\sigma_e \ll q$. The hardness of LWE is related to the hardness of well-known lattice problems such as the (approximate) Shortest Vector Problem (SVP).

**Most classical attacks on LWE** rely on lattice reduction [41, 49, 19]. For example, given $m$ samples $(\mathbf{a_i}, b_i)$, create a matrix $A_{m \times n}$ whose $m$ rows are the vectors $\mathbf{a_i}$. The unique shortest vector problem (uSVP) attack recovers the secret $\mathbf{s}$ by finding the shortest vector in a lattice constructed from $\mathbf{b}$, the columns of $A_{m \times n}$, and other parameters. The best known algorithms for solving SVP run in time exponential in the dimension $n$. Somewhat counter-intuitively, the smaller the modulus $q$, the harder the LWE problem. Approximate solutions can be computed in polynomial time with LLL [41], but the approximation factor is exponentially bad (exponential in $n$). We compare our results with uSVP attacks on LWE in §A.7.

**Machine learning (ML) attacks on LWE.** SALSA [58], the seminal ML attack on LWE, uses a large collection of LWE samples $\{(\mathbf{a}, b)\}$ with the same secret $\mathbf{s}$ to train a transformer [57] that predicts $b$ from $\mathbf{a}$. SALSA presents methods for secret recovery via queries to the trained model, and observes that high model accuracy is not needed: secrets are recovered as soon as the transformer *starts* to learn (training loss drops). SALSA is a proof of concept, recovering binary secrets with 3 or 4 nonzero bits for problems with dimension up to 128, small instances of LWE solvable via exhaustive search.

**PICANTE** builds on an observation from SALSA ([58, Table 4]): transformers trained on LWE samples $\{(\mathbf{a}, b)\}$, with entries of $\mathbf{a}$ drawn from a restricted range instead of all of $[0, q)$, can recover binary secrets with larger Hamming weights $h$ (number of nonzero bits). So PICANTE introduces a data preprocessing phase during which the LWE samples $(\mathbf{a}, b)$ are processed by BKZ, a lattice reduction algorithm, to obtain LWE samples with the same secret but smaller coordinate variance (and larger error, see §2). In addition to training the transformer on the preprocessed samples, PICANTE reduces the number of LWE samples required for the attack from 4 million in SALSA to $4n$ (e.g. 1400 for $n = 350$), and improves secret recovery. Overall, PICANTE can recover binary secrets for dimensions up to 350 and Hamming weight up to 60. This is a considerable improvement over SALSA, faster than the uSVP attacks we compare against, and out of reach for exhaustive search.

**PICANTE has several limitations**. First, it only recovers sparse binary secrets, an important but limited subclass of LWE. Homomorphic encryption (HE) may use binary secrets, but HE and other PQC schemes typically use ternary ($s_i \in \{-1, 0, 1\}$) or small ($|s_i| < k$, $k$ small) secrets. Second, PICANTE's preprocessing is costly as dimension increases, making it difficult to scale PICANTE to dimensions larger than 350. For $n = 512$, $\log_2 q = 45$, PICANTE's preprocessing approach could not finish in a month with full parallelization. Third, PICANTE only experiments with large modulus $q$: $\log_2 q = 23$ for $n = 256$, $\log_2 q = 27$ for $n = 300$, and $\log_2 q = 32$ for $n = 350$. Practical LWE-based systems use small $q$: LIZARD [21] recommends $\log_2 q = 10$ for $n = 608$ with sparse binary secrets, and the HE standard [4] recommends $\log_2 q = 25$ for $n = 1024$ with ternary secrets.

**Our work, SALSA VERDE**, improves on PICANTE and makes the following contributions:

- We introduce a **two-bit distinguisher, a new secret recovery technique** for sparse binary, ternary and small Gaussian secrets. VERDE fully recovers binary and ternary secrets equally well (§3).
- We improve data **preprocessing techniques**, making them forty times faster and 20% more effective, enabling recovery of binary, ternary and small Gaussian secrets for dimension 512 (§4).
- We **decrease the modulus** $q$, showing VERDE outperforming uSVP attacks (§5 and §A.7).
- We propose **NoMod**, a framework for understanding the success of ML-based LWE attacks (§5).
- We present a **theoretical analysis** to show heuristically that successful secret recovery depends only on $\sqrt{h}$ and the standard deviation of the distribution of the LWE data (§6).
- We experiment with **encoder-only models** and compare their success with seq2seq models (§7).

**Key results.** Our main finding is that small, sparse LWE secrets are weak. For dimension $n = 256$, we can recover binary and ternary secrets with 10% sparsity ($h/n$) for $\log_2 q = 20$, and with 3% sparsity when $\log_2 q = 12$ (Table 1). For $n = 512$, $\log_2 q = 41$, we recover binary and ternary secrets with $\geq 11\%$ sparsity. Furthermore, VERDE scales well to higher dimensions for small sparse secret recovery. Training VERDE models on $n = 256/350/512$ problems takes only 1.5/1.6/2.5 hours per epoch, a small proportional increase compared to the increase in $n$. Also, we find that VERDE runs faster than uSVP attacks, at the expense of using more compute resources in parallel (see A.7).

In Table 1, we report the timings for successful secret recoveries, for binary/ternary/Gaussian secrets, with varying $n$ and $\log_q$. The table records the highest $h$ recovered for each column: for binary secrets,

$h$ is the Hamming weight, and for ternary and Gaussian secrets, $h$ is the number of non-zero entries. We record the amount of time needed for each stage of the attack, preprocessing (in hours/CPU, assuming full parallelization), model training (hours/epoch · (# epochs)), and total attack time. For full parallelization, the number of CPU cores required is 4 million divided by $2n$. Source code and parameters to reproduce our main experiments are included in the supplementary material. The full code base will be open-sourced.

**Table 1.** VERDE *attack times (preprocessing, model training, and total), for dimension $n$ and $\log_2 q$.* $h$ *= # non-zero entries in recovered secrets, $b$ = binary, $t$ = ternary, $g$ = Gaussian secret distributions.*

| $(n, \log_2 q)$ | (256, 12) | | | (256, 20) | | | (350, 21) | | | (350, 27) | | | (512, 41) | | |
|---|---|---|---|---|---|---|---|---|---|---|---|---|---|---|---|
| secret distribution | b | t | g | b | t | g | b | t | g | b | t | g | b | t | g |
| highest $h$ | 8 | 9 | 5 | 33 | 24 | 7 | 12 | 13 | 5 | 36 | 36 | 10 | 63 | 58 | 16 |
| preprocessing (hrs/CPU) | 1.5 | 1.5 | 1.5 | 7.5 | 7.5 | 7.5 | 16 | 16 | 16 | 216 | 216 | 216 | 840 | 840 | 840 |
| training time (hrs) | 1.5 | 3 | 12 | 3 | 7.5 | 1.5 | 1.6 | 25.6 | 1.6 | 1.6 | 17.6 | 3.2 | 17.5 | 27.5 | 2.5 |
| total time (hrs) | 3 | 4.5 | 13.5 | 10.5 | 15 | 9 | 17.6 | 41.6 | 17.6 | 218 | 234 | 220 | 858 | 868 | 843 |

**Scope of results.** Instances of cryptographic problems like LWE can be broadly categorized as easy (solvable via exhaustive search), medium-to-hard (requiring significant resources to solve), or standardized (believed secure). VERDE attacks *medium-to-hard* LWE problems (parameterized by dimension $n$, Hamming weight $h$, modulus $q$). VERDE does not attack toy problems (like SALSA did), nor does it attack the NIST standard directly. Rather, VERDE demonstrates successful attacks on medium-to-hard LWE problems using tools from AI, improving our understanding of the security of proposed LWE-based cryptosystems.

## 2 SALSA VERDE Overview

In this section we describe the SALSA VERDE attack and relevant parts of its predecessor PICANTE.

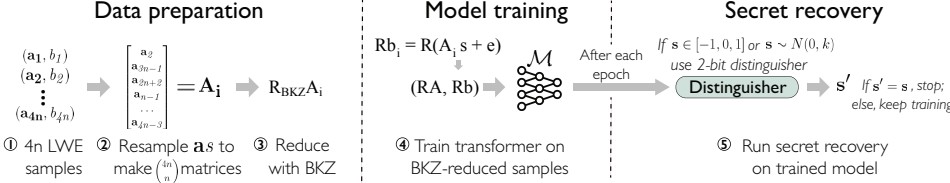

**Figure 1.** *Overview of VERDE's attack methodology*

**High-level overview.** Like PICANTE, VERDE starts with $4n$ LWE samples with the same secret $\mathbf{s}$. In practice, this data would be eavesdropped. VERDE then proceeds in three stages: preprocessing, model training and secret recovery (see Figure 1). The preprocessing stage augments the $4n$ initial $(\mathbf{a}, b)$ pairs to 2 million, then runs lattice reduction to yield a training set of 4 million samples with the same secret $\mathbf{s}$. The preprocessed data is used to train a transformer to predict $b$ from $\mathbf{a}$. After each training epoch (2 million LWE samples), the model is queried to form a secret guess. The attack succeeds if the secret guess is correct, tested via statistical methods without knowledge of the actual secret. Otherwise, the model is trained for another epoch, and secret recovery is run again.

**Data preprocessing.** VERDE's preprocessing is similar to PICANTE's, with several improvements. First, we create $n \times n$ matrices $\mathbf{A}_i$ by sampling without replacement $n$ of the $4n$ original LWE samples. Then, we apply lattice reduction to the matrices $\mathbf{A}_i$ to reduce the standard deviation of their entries (initially uniformly distributed over $[0, q)$). This process generates $2n$ preprocessed samples $(\mathbf{a}', b')$, with the same secret and is implemented in parallel to create a training set of 4 million samples.

During lattice reduction, PICANTE applies BKZ (as implemented in *fplll* [27]) to the $2n \times 2n$ matrix: $\mathbf{\Lambda}_i = \begin{bmatrix} \omega \cdot \mathbf{I}_n & \mathbf{A}_i \\ 0 & q \cdot \mathbf{I}_n \end{bmatrix}$. BKZ finds a linear transformation $[\mathbf{R}_i \quad \mathbf{C}_i]$ such that the norms of the $2n$ rows of $[\mathbf{R}_i \quad \mathbf{C}_i] \mathbf{\Lambda}_i = [\omega \cdot \mathbf{R}_i \quad \mathbf{R}_i \mathbf{A}_i + q \cdot \mathbf{C}_i]$ are small. Applying $\mathbf{R}_i$ to $\mathbf{A}_i$ and $\mathbf{b}_i$, PICANTE generates $2n$ reduced LWE pairs $(\mathbf{R}_i \mathbf{A}_i, \mathbf{R}_i \mathbf{b}_i)$ (modulo $q$). VERDE instead rearranges the rows of $\mathbf{\Lambda}_i$ and applies lattice reduction to $\mathbf{\Lambda}'_i = \begin{bmatrix} 0 & q \cdot \mathbf{I}_n \\ \omega \cdot \mathbf{I}_n & \mathbf{A}_i \end{bmatrix}$. This reduces the number of operations needed for lattice reduction and allows BKZ to run with lower floating point precision. These two improvements cut the preprocessing time significantly.

They also result in smaller $\mathbf{R}_i$ norms (smaller error in $\mathbf{R}_i\mathbf{b}_i$), which can be leveraged to further reduce the norm of $\mathbf{R_i A_i} + q \cdot \mathbf{C_i}$ by using a lower $\omega$ parameter ($\omega = 10$) than PICANTE ($\omega = 15$). Finally, VERDE replaces BKZ with two interleaved algorithms (BKZ 2.0 [19] and the efficient reduction technique introduced in [15]), adaptively increases the blocksize and precision as reduction progresses, and introduces a stopping condition. For instance, for $n = 256$ and $\log_2 q = 20$, VERDE's preprocessing is $45\times$ faster, while improving the quality of the reduction by $20\%$ (Table 2).

**Table 2.** *Impact of successive improvements to preprocessing*. $n = 256$, $\log_2 q = 20$. *Reduction factor of the standard deviation of entries of* $\mathbf{A}_i$: *lower is better. Time: # of hours to preprocess one matrix on one CPU.*

| preprocessing technique | reduction factor | time (hrs/CPU) |
|---|---|---|
| PICANTE | 0.53 | 338 |
| + reordered $\mathbf{\Lambda}_i$ rows | 0.47 | 136 |
| + reduced floating point precision | 0.47 | 24 |
| + reduced parameter $\omega$ from 15 to 10 | 0.43 | 38 |
| VERDE (+ interleaved reduction, adaptive blocksize, early stopping) | **0.43** | **7.5** |

Quality of reduction is measured by the standard deviation of the entries of $\mathbf{A}_i$. Preprocessing time in Table 2 is the hours needed to process one matrix on a single CPU. We list the time for all $n, q$ attempted in Tables 5 and 19. For each dimension $n$ and modulus $q$, we process 2 million/$n$ matrices in parallel across hundreds of CPUs, see §A.1 for details. Better preprocessing allows VERDE to scale to larger dimensions and retrieve secrets with larger $h$.

**Transformer training.** The 4 million reduced LWE pairs are used to train a transformer to predict $b$ from $\mathbf{a}$. The values $b$ and the coordinates of $\mathbf{a}$ are integers in $[0, q)$. They are represented in base $B = \lceil q/8 \rceil$ (for $\log_2 q > 30$, $B = \lceil q/16 \rceil$) and encoded as sequences of two tokens over a vocabulary of $2,000$ (see §A.1 for a discussion of these choices). Model training is framed as a translation task, from a sequence of $2n$ tokens representing $\mathbf{a}$ to a sequence of 2 tokens representing $b$ (see [40, 12] for similar uses of transformers for mathematical calculations). The model is trained to minimize the cross-entropy between model prediction and the sequence of tokens representing $b$, using the Adam optimizer with warmup [39] and a learning rate of $10^{-5}$. For $n = 256, 350$ and $512$, each epoch uses 2 million LWE samples and runs for $1.5, 1.6,$ or $2.5$ hours. Time/epoch doesn't vary with $q$ or secret type. Our models train on one NVIDIA V100 32GB GPU and often succeed in the first epoch for low $h$. Number of epochs required are included in many tables throughout, including Tables 3 and 4.

VERDE uses the same architecture as PICANTE: a sequence to sequence (seq2seq) transformer [57], with a one-layer encoder (dimension 1024, 4 attention heads), and a 9-layer decoder (dimension 512, 4 heads). The last 8 layers of the decoder are shared (i.e. they form a Universal Transformer [25]). Iteration through shared loops is controlled by the copy-gate mechanism introduced in [22].

Seq2seq models allow output sequences to be longer than inputs, a useful feature for machine translation but not necessary in our setting. For comparison, we also implement a simpler *encoder-only* transformer, that is 4-layer BERT-like (dimension 512, 4 heads), together with rotary word embeddings (analogous to the rotary position embeddings [55]) to account for the modular nature of the problem. On top of this, we also add an earth mover's distance (EMD) auxiliary objective. We compare this model's performance to that of the seq2seq model (§7).

**Secret recovery.** Secret recovery runs after each epoch (2 million LWE samples). PICANTE used three recovery methods: direct recovery, cross-attention and distinguisher. Direct recovery struggles as dimension increases, because it relies on accurate model evaluations at special $\mathbf{a}$ values which are out of distribution for the training set. Cross-attention is incompatible with encoder-only architectures and is consistently outperformed by the distinguisher on VERDE-preprocessed data. Thus, VERDE only uses the distinguisher, which works as follows: for any test vector $\mathbf{a}_{\text{test}}$ and random $K$, if the $i$-th entry of the secret $s_i$ is zero, then $(\mathbf{a}_{\text{test}} + K\mathbf{e}_i) \cdot \mathbf{s} = \mathbf{a}_{\text{test}} \cdot \mathbf{s}$ (where $\mathbf{e}_i$ is the $i$-th standard basis vector). Therefore, for each $i$, the distinguisher computes the difference between model predictions on $\mathbf{a}_{\text{test}}$ and $\mathbf{a}_{\text{test}} + K\mathbf{e}_i$ for different $\mathbf{a}_{\text{test}}$. If differences are small, the corresponding $s_i$ is likely zero. For ternary and Gaussian secrets, the distinguisher is modified (see §3).

For successful secret recovery, the trained model must generalize well to $\{\mathbf{a}_{\text{test}}\}$, the vectors used for testing. In PICANTE, the distinguisher runs on random $\mathbf{a}_{\text{test}}$, with coordinates uniform in $[0, q)$. However, the model is trained on preprocessed $\mathbf{a}_{\text{train}}$, with a non-uniform coordinate distribution. So PICANTE's distinguisher recovery requires that the model generalize outside its training distribution.

This is known to be a difficult ML task. Instead, VERDE runs the distinguisher on a held-out subset of 128 preprocessed vectors $\mathbf{a}_{\text{test}}$. Since the test vectors have the same distribution as the training set, the trained model only needs to generalize in-distribution, a much easier task. This change in $\mathbf{a}_{\text{test}}$ improves the performance of the distinguisher (see Table 14 in §A.4).

In practice, for each secret coordinate, the distinguisher computes the sum of the absolute difference between model predictions at $\mathbf{a}_{\text{test}}$ and $\mathbf{a}_{\text{test}} + K\mathbf{e}_i$, for 128 vectors $\mathbf{a}_{\text{test}}$ and a random $K \in (0.3q, 0.7q)$ for each $\mathbf{a}_{\text{test}}$. The model makes a secret prediction $\mathbf{s}'$ by setting the bits to 1 on the $h$ coordinates with the largest sums, and verifies $\mathbf{s}'$ by computing $\mathbf{a} \cdot \mathbf{s}' - b$ on the original $4n$ LWE samples. If the secret is correctly predicted, this quantity should always be small. The statistical properties of this test are discussed in section A.2 of [42].

Having discussed VERDE's background and described its methodology, we now present VERDE's key results, as summarized in §1. We begin with results on ternary and narrow Gaussian secrets.

## 3 Secret Distributions

One of VERDE's major contributions is a method to recover secrets from sparse ternary and narrow Gaussian distributions, both of which are being considered for use in real-world cryptosystems [4]. Prior ML-based LWE attacks (PICANTE and SALSA) only recovered sparse binary secrets. Here, we describe VERDE's method for recovering these more general secret distributions, and its performance on ternary and small Gaussian secrets for fixed dimension $n = 256$ and $\log_2 q = 20$. Throughout this paper, $h$ denotes the number of nonzero entries in a secret, which is equal to the Hamming weight in the binary case. We define secret *sparsity* as the percentage of nonzero secret entries $h/n$.

**Recovering ternary and small Gaussian secrets.** In lattice-based cryptography, ternary secrets are vectors $\mathbf{s}$ of dimension $n$, with entries equal to 0, 1, or $-1$ (with equal probability of 1 and $-1$ in sparse secrets). Gaussian secrets are vectors $\mathbf{s}$ of dimension $n$, with entries drawn from a Gaussian distribution with small standard deviation $\sigma$. In this paper, we use $\sigma = 3$. The Homomorphic Encryption Standard [4] includes secure parameter choices for Gaussian secrets with $\sigma = 3.2$.

Ternary and Gaussian secrets introduce new challenges for ML-based attacks on LWE. The recovery methods in PICANTE distinguish zero from nonzero bits. In the binary case, this produces one guess $\mathbf{s}_{\text{guess}}$, which can be verified by checking that $b - \mathbf{a} \cdot \mathbf{s}_{\text{guess}}$ is small. In the ternary case, PICANTE would produce $2^h$ secret guesses (about $12^h$ guesses for Gaussian secrets), due to the additional $-1$ entries, and verification becomes very expensive as $h$ increases. VERDE recovers ternary secrets using a two-step approach: first, *partial recovery* distinguishes zero from nonzero entries, then *full recovery* guesses the sign of the nonzero bits.

*Partial recovery.* To identify nonzero bits, VERDE uses the binary secret distinguisher from §2 (after [42, Section 4.3]). For each secret bit, it computes a score from a sample of reduced LWE pairs. The $h$ bits with the highest scores are candidate nonzero bits. VERDE assumes $h$ is not known and runs the next step for all reasonable possible values of $h$, e.g. from 1 to $n/20$. Partial recovery alone is a major contribution, as we are not aware of existing attacks that can identify nonzero secret bits.

*Full recovery (ternary secrets).* To determine whether nonzero bits of a ternary secret are 1 or $-1$, VERDE introduces a novel two-bit distinguisher, leveraging the following observation. If two nonzero secret bits $s_i$ and $s_j$ are equal, then for any $\mathbf{a}$, exchanging the coordinates $a_i$ and $a_j$ will result in the same $b$, and corresponding model predictions will be close. Otherwise, model predictions will be different (if $s_i \neq s_j$). Similarly, if $s_i = s_j$, for any $\mathbf{a}$ and $c \neq 0$, changing $a_i \rightarrow a_i + c$ and $a_j \rightarrow a_j - c$ yields the same $b$, and close model predictions. The two-bit distinguisher uses these techniques to compare each nonzero bit with all others, therefore defining two classes of nonzero bits. Letting one class of bits be 1 or $-1$, VERDE produces two secret guesses to be verified.

*Full recovery (Gaussian secrets).* At present, we implemented full recovery only for binary and ternary secrets. However, full recovery of small Gaussian secrets is possible via the following adaptation of the two-bit distinguisher. The two-bit distinguisher groups nonzero secret bits into $k$ classes believed to have the same value. In our case, the nonzero bits follow a Gaussian distribution with $\sigma = 3$, so we may safely assume that all non-zero secret bits are in $[-9, 9]$ (within 3 standard deviations) – i.e. $k = 18$. Since the secret is Gaussian, we expect the largest classes to correspond to the values $-1$ and 1, followed by $-2$ and 2, and so on. Therefore, we can intelligently assign values to classes based on class size, and test the corresponding $2^{k/2} = 512$ secrets. We leave

implementation of this as future work and report the performance of partial Gaussian secret recovery, using knowledge of $\mathbf{s}$ to validate correctness.

**Table 3.** ***Partial and full ternary secret recovery.*** $n = 256, \log_2 q = 20$. *Epoch when secret is recovered.*

| $h$ | 5 | 10 | 15 | 20 | 21 | 22 | 23 | 24 | 25 |
|---|---|---|---|---|---|---|---|---|---|
| partial recovery | 8/10 | 6/10 | 6/10 | 2/10 | 3/10 | 2/10 | 1/10 | 3/10 | 1/10 |
| training epoch | 0,0,0,0,0,0,1,7 | 0,0,1,1,1,1 | 0,0,0,1,2,8 | 0,2 | 0,1,3 | 2,5 | 9 | 0,4,7 | 2 |
| full recovery | 8/10 | 6/10 | 5/10 | 1/10 | 3/10 | 2/10 | 0/10 | 1/10 | 0/10 |
| training epoch | 0,0,0,0,0,0,1,7 | 1,2,2,6,7,8 | 1,1,6,9,11 | 5 | 4,10,17 | 8,22 | | 5,12 | |

**Table 4.** ***Partial Gaussian secret recovery.*** $n = 350, \log_2 q = 27$. *Epoch when secret is recovered.*

| $h$ | 4 | 5 | 6 | 7 | 8 | 9 | 10 |
|---|---|---|---|---|---|---|---|
| partial recovery | 8/10 | 8/10 | 7/10 | 5/10 | 2/10 | 2/10 | 4/10 |
| training epoch | 0,0,0,0,1,1,1,1 | 0,0,0,1,2,3,3,7 | 0,0,1,2,2,5,10 | 2,2,3,7,10 | 0,2 | 1,9 | 1,5,6,12 |

**VERDE's performance across secret distributions.**
VERDE recovers *ternary secrets* with sparsity up to 10%, with comparable performance on binary secrets. Table 3 provides details on partial and full ternary secret recovery for $n = 256$ and $\log_2 q = 20$ and $h = 5 - -25$. For low values of $h$ ($h < 20$), ternary secrets are partially recovered early during training (i.e. mostly in epoch 0 or 1, during the first pass on preprocessed data), and usually fully recovered in the same epoch or shortly after. As $h$ increases, more training is required for recovery, and the delay between partial and full recovery increases.

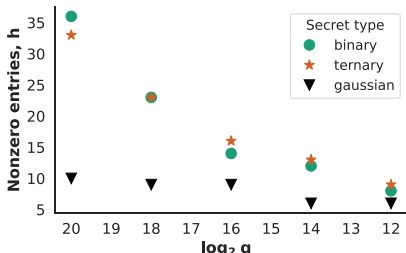

**Figure 2.** ***Best $h$ recovered vs.*** $\log_2 q$ ***and secret distribution.*** $n = 256$.

For *small Gaussian secrets*, VERDE only implements partial recovery (recovery of the nonzero bits). Table 4 presents results for $n = 350$ and $\log_2 q = 27$. Recovered $h$ are lower than in the binary case, and models require less training for low $h$.

Figure 2 compares VERDE's performance across secret distributions for problems with $n = 256$ and different moduli $q$. For each setting, we run 100 recovery experiments and report the highest $h$ secret recovered in those attempts. Recovery is comparable for binary and ternary secrets. Small Gaussian secrets are significantly harder.

## 4  Large Dimension

The hardness of LWE increases as $n$ grows. PICANTE recovered sparse binary secrets for dimension up to $n = 350$. VERDE pushes this limit to $n = 512$, for sparse binary, ternary, and narrow Gaussian secrets. For $n = 512$ and $\log_2 q = 41$, VERDE recovers binary and ternary secrets with sparsity up to 0.12 (highest $h = 63, 60$) and Gaussian secrets with $h$ up to 16. The longer VERDE's preprocessing step runs, the higher $h$ secrets it can recover. Recall that real-world schemes like LIZARD operate in dimension $n = 608$ [21] and HE in dimension $n = 1024$ [4]. Those systems use significantly smaller $q$ where LWE is harder: $\log_2 q = 10$ and $27$, respectively, compared to VERDE's $\log_2 q = 41$.

**VERDE's performance in large dimension.** Table 5 shows VERDE's performance for dimension $n = 512$, after 7 to 35 days of preprocessing, on the same set of matrices $\mathbf{A}_i$. For each preprocessing time, we measure the quality of lattice reduction via a "reduction factor," computed by taking the ratio of the standard deviation of reduced $\mathbf{A}_i$ entries to the standard deviation of a random matrix with uniform coefficients $0 \le a_i < q$, i.e. $\frac{\text{stddev}(\mathbf{A}_{\text{bkz}})}{\text{stddev}(\mathbf{A}_{\text{rand}})}$, with $\text{stddev}(\mathbf{A}_{\text{rand}}) = \frac{q}{\sqrt{12}}$. This metric, used in PICANTE for selecting BKZ parameters, is discussed in § 6. As Table 5 demonstrates, the maximum recoverable $h$ is strongly correlated to the quality of lattice reduction.

**Preprocessing adjustments for large $n$.** Scaling up to $n = 512$ requires a number of adjustments to our preprocessing methodology. For $n = 512$, the first loops in BKZ 2.0 are very slow. To avoid this, we use BKZ with smaller blocksizes than those used for $n = 256$ and $350$ (see §A.1). Also, lattice

**Table 5.** ***Data preprocessing vs performance***. $n = 512$, $\log_2 q = 41$. *Highest values of $h$ recovered, for different reduction factors (lower factor = better reduction).*

| preprocess time | reduction factor | binary $h$ | ternary $h$ | Gaussian $h$ |
|---|---|---|---|---|
| 7 days | 0.519 | 16,17,17,20 | 17,20,20,21,21 | 8,8,8,8,10,10,13 |
| 10 days | 0.469 | 21,22,23,28 | 22,24,24,27,27,29 | 11,11,11,12,12,12 |
| 14 days | 0.423 | 32,32,34,34,35,40 | 32,34,34,35,35 | 11,11,11,12,12,12 |
| 20 days | 0.380 | 35,35,36,41,49 | 35,35,37,45,46 | 13,13,13,16 |
| 28 days | 0.343 | 40,43,45,47,50,51,55 | 40,41,41,44,45,48,48,53 | 13,13,14,16,16 |
| 35 days | 0.323 | 48,48,49,52,57,59,63 | 45,46,50,55,58,60 | 14,16 |

reduction is significantly slower for larger matrices. To mitigate this, for $n = 512$, we use $448$ LWE samples (instead of $512$) when generating $\mathbf{A}_i$ for lattice reduction, therefore reducing the matrix size from $1024 \times 1024$ to $960 \times 960$. Experimentally, we observe that using slightly fewer samples did not negatively impact our reduction factor or attack performance.

## 5 Small Modulus

To define real-world parameters for lattice-based cryptography, standardization committees and communities (e.g. [4, 17]) select a small enough modulus $q$ (for fixed dimension $n$), so that all known attacks are (heuristically) predicted to run in time at least $2^{128}$, therefore attaining the U.S. government minimum 128-bit security level. For classical lattice reduction attacks, the smaller the modulus, the more difficult the attack. This is because lattice reduction algorithms such as LLL and BKZ attempt to compute short vectors in Euclidean space, but cryptosystems operate modulo $q$. Smaller moduli result in smaller lattice volumes, meaning shorter target vectors are required to break the system. In our ML approach, we also observe that VERDE is less likely to succeed when $q$ is smaller (see Table 6). Nevertheless, VERDE outperforms the uSVP attack (§A.7).

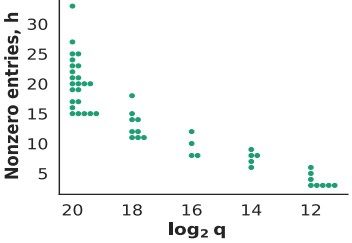

**Figure 3.** $h$ ***of recovered secrets vs.*** $\log_2 q$, $n = 256$. *10 random binary secrets attempted for each $h$. One green dot represents a successful recovery.*

**Table 6.** ***Highest*** $h$ ***recovered,*** $n = 256$, $350$. *Secret distributions are b = binary, t = ternary, g = Gaussian.*

| $n, \log_2 q$ | reduction factor | recovered $h$ | | |
|---|---|---|---|---|
| | | b | t | g |
| 256, 20 | 0.43 | 33 | 24 | 7 |
| 256, 18 | 0.53 | 18 | 19 | 7 |
| 256, 16 | 0.63 | 12 | 12 | 6 |
| 256, 14 | 0.71 | 9 | 9 | 6 |
| 256, 12 | 0.77 | 6 | 6 | 5 |
| 350, 27 | 0.38 | 36 | 36 | 10 |
| 350, 21 | 0.61 | 12 | 13 | 5 |

**PICANTE vs. VERDE's performance on small** $q$. PICANTE attacks larger moduli: it can recover binary secrets with $h = 31$ for $\log_2 q = 23$ and $n = 256$, and $h = 60$ for $\log_2 q = 32$ and $n = 350$. Table 6 presents VERDE's highest recovered $h$ (in 10 random attempts) for dimensions 256 and 350 and different $q$ for binary, ternary, and narrow Gaussian secrets. First, note that VERDE recovers binary secrets with $h = 33$ for $n = 256$ and $\log_2 q = 20$, but also, VERDE succeeds for much smaller $q$, as small as $\log_2 q = 12$, a near-real-world parameter setting, demonstrating that VERDE significantly outperforms PICANTE. However, for smaller $q$, VERDE recovers only secrets with smaller $h$. As with classical attacks, the likely culprit is lattice reduction: the reduction factor after preprocessing is $0.71$ for $\log_2 q = 12$ versus $0.43$ for $\log_2 q = 20$, and $h$ for recovered binary secrets drops from 33 to 6. Section 6 provides a theoretical explanation of this phenomenon.

Figure 3 visualizes VERDE's success rates for $n = 256$ with binary secrets. For every value of $q$ and $h$, we run VERDE on 10 binary secrets, using the same preprocessed data. VERDE's success rate decreases as $h$ increases. Attempting 10 random binary secrets for $n = 256$, $\log_2 q = 12$, VERDE recovers secrets for up to $h = 6$ but not $h = 7, 8$ (Table 6). However, with more attempts, VERDE recovers $1/100$ binary secrets for $h = 8$. Experiments with different random seeds (see §A.2) suggest that model initialization alone is not responsible for how success rate trends with $h$.

**Explaining success for smaller** $q$ **via NoMod.** As Table 6 indicates, for given $n$ and $h$, secrets are harder to recover for smaller $q$. This suggests that the modular operations in the computation of $b$

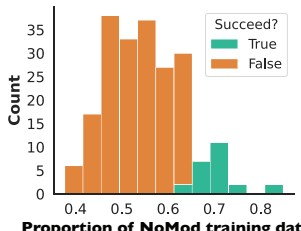

**Figure 4.** *Effect of NoMod data on secret recovery for* $n = 256$, *binary secrets.* Count = # of experiments.

**Table 7.** **NoMod** % *before/after permuting the columns of* $A$. $n = 256, \log_2 q = 14$, *binary secrets. Each column is a random secret for each* $h = 6, 7, 8$. *Entries are the* **NoMod** %. **Green** = *secret recovered;* *red or black* = *failure.*

| h | method | NoMod percentages and recovery success | | | | | | | | | |
|---|---|---|---|---|---|---|---|---|---|---|---|
| 6 | original | 56 | 61 | 60 | 61 | 56 | *52* | 67 | 67 | **76** | 67 |
|   | permuted | 57 | 67 | 56 | 67 | 62 | **67** | 60 | 60 | *52* | 56 |
| 7 | original | *60* | 60 | 52 | *49* | 60 | **75** | 55 | *60* | 55 | 56 |
|   | permuted | **67** | 52 | 60 | **75** | 60 | **73** | 60 | **71** | 59 | 66 |
| 8 | original | 60 | **74** | **74** | 66 | 60 | 66 | 55 | 60 | 63 | 55 |
|   | permuted | 55 | *55* | *60* | 52 | 55 | 60 | 55 | 49 | 60 | 60 |

from $\mathbf{a}$ might account for the difficulty. To investigate this, we evaluate, for a given known binary secret $\mathbf{s}$, the percentage of samples where computing $b$ did not require the modular operation, for the 4 million samples in our training set. More precisely, we represent the mod $q$ coordinates of $\mathbf{a}$ and $b$ in the interval $(-q/2, q/2)$, and compute $x = \mathbf{a} \cdot \mathbf{s} - b$ without modular operations. If $b$ was computed without modular operations, then $x$ is equal to the error for that sample, which is small. Otherwise $x$ is equal to the error plus a multiple of $q$. For each $(\mathbf{a}, b)$, if $|x| < q/2$, then no modular operation was performed. We define **NoMod** to be the percentage of such $x$ in the training set.

Figure 4 shows the distribution of secret recoveries, for varying **NoMod**, for 210 experiments with dimension $n = 256$ and $\log_2 q = 20$. Clearly, recovery occurs when **NoMod** exceeds a threshold value, empirically observed to be $67\%$. These results confirm our hypothesis that modular arithmetic is the major cause of failure to recover secrets, and help explain the observation ([42, §6.4] and §A.2) that some secrets are never recovered, no matter how many model initializations are tried. For a given secret, **NoMod** is a property of the training data, not of model initialization, and multiple initializations only seem to help when **NoMod** is close to the recovery threshold.

**A trick for improving attack success.** The **NoMod** percentage can only be calculated if the secret is known, so it cannot aid real-world attack settings. However, our **NoMod** observations suggest that, if recovery fails for a given secret $\mathbf{s}$, the failure may be due to a low **NoMod** factor in the preprocessed training set. This reveals a strategy for potentially recovering such secrets. If an initial run of VERDE fails for a secret $\mathbf{s}$, start over with the (un-preprocessed) matrices $\mathbf{A}_i$, sampled from the original LWE samples. For each of these, apply a random permutation $\Pi$ to the columns and preprocess the permuted matrices. This creates a new dataset of $\mathbf{A}'$ with corresponding $b'$, associated to a permuted counterpart $\mathbf{s}'$ of the original secret $\mathbf{s}$. If the **NoMod** of $\mathbf{s}'$, $b'$ and $\mathbf{A}'$ is higher that that of $\mathbf{s}$, $b$ and $\mathbf{A}$ (though this cannot be measured), $\mathbf{s}'$ can be recovered. If the attack succeeds, $\mathbf{s}$ can be restored by applying the inverse of permutation $\Pi$. Table 7 presents the impact of permutations on **NoMod** for 10 secrets and $h = 6, 7, 8$ for $n = 256$ and $\log_2 q = 14$. Some secrets become recoverable after using the permutation trick.

## 6 A Theoretical Scaling Law for VERDE

**Intuition.** The **NoMod** experiments provide a key insight about the hardness of secret recovery in SALSA-based attacks on LWE. They suggest that a secret $\mathbf{s}$ can be recovered from a training set if over $67\%$ of the $\{x = \mathbf{a} \cdot \mathbf{s} - b\}$ are concentrated in the interval of length $q$. If the random variable $x$ is Gaussian (a reasonable assumption for $h \gg 1$, since the entries of $\mathbf{a}$ are random and bounded), $68\%$ of its values will be within one standard deviation of its mean, i.e. spread over two standard deviations. Therefore, for the secret to be recoverable, the standard deviation of $x$ should satisfy $\sigma_x \leq q/2$. If $\mathbf{s}$ is a binary secret with Hamming weight $h$, and the entries of $\mathbf{a}$ have standard deviation $\sigma_a$, we have $x \approx \mathbf{a} \cdot \mathbf{s} - e$ and $\sigma_x \approx \sqrt{h}\sigma_a + \sigma_e \approx \sqrt{h}\sigma_a$. Therefore, $\mathbf{s}$ is recoverable if $\sqrt{h}\sigma_a \leq q/2$, or $\sigma_a \leq \frac{q}{2\sqrt{h}}$.

**Scaling Laws.** We now apply this insight to the experimental results of ML-based attacks on LWE. Consider the original SALSA attack, which does not utilize data preprocessing. Since the entries of $\mathbf{a}$ are uniformly distributed over $[0, q)$, $\sigma_a = \frac{q}{\sqrt{12}}$. Replacing in $\sigma_a \leq \frac{q}{2\sqrt{h}}$ yields $h \leq 3$, the main experimental result of SALSA. If we constrain the entries of $\mathbf{a}$ to be in $[0, \alpha q)$ (Table 4 in [58]), we have $\sigma_a = \frac{\alpha q}{\sqrt{12}}$, and $h \leq \frac{3}{\alpha^2}$. Applying this formula for $\alpha \in 0.6, 0.55, 0.5, 0.45$, we obtain maximal recoverable Hamming weights of 8, 10, 12 and 15, which closely match SALSA experimental results.

These results shed light on the role of preprocessing in PICANTE and VERDE. When the standard deviation of **a** is reduced by a factor $\alpha$, maximal recoverable $h$ increases by a factor $\frac{1}{\alpha^2}$. However, the formula $h \leq \frac{3}{\alpha^2}$ underestimates actually recovered $h$ by a factor of 2. For instance, from the reduction factors from Table 5, we should expect recovered $h$ to range from 11 to 29 as preprocessing time increases from 7 to 35 days, but actual recovered $h$ ranges from 20 to 63. As seen in §5, the preprocessing step makes some secrets easier to recover for the same $h$, so VERDE performs better than predicted by theory for some secrets (at the expense of other secrets). Finally, note that the formula for the standard deviation of $x$ is the same for ternary and binary secrets. This accounts for the observation in §3 that ternary secrets are of similar difficulty for VERDE as binary secrets.

## 7 Model architecture

VERDE's baseline model uses a seq2seq architecture as in SALSA and PICANTE. Here, we compare it to the new encoder-only model discussed in §2. Specifically, this new model is based on the DeBERTa [34] model but replaces its "relative positional embeddings" with rotary word embeddings, in the spirit of rotary positional embeddings (RoPE) [55] but applied to the integer words. It has 4 layers, 4 heads and 512 dimensions. On top of the cross-entropy loss, we add an auxiliary, squared earth mover's distance (EMD) loss to compare model's softmax distribution with the target $b$. This encourages the model to make predictions that are at least close to the targets, if not exact matches. Trained with the auxiliary EMD loss, the model also replaces the beam search used in the distinguisher with a novel EMD-based distinguisher that compares the difference between the *distributions* produced by the model on $\mathbf{a}_{\text{test}}$ and $\mathbf{a}_{\text{test}} + K\mathbf{e}_i$.

**Table 8.** *Performances of seq2seq and encoder-only models.* $n = 256, \log_2 q = 12; n = 512, \log_2 q = 41.$ *For binary and ternary secrets, we run 10 secrets per $h$ and indicate the epochs of full recovery.*

| $n$, secret $\chi_s$ | 256, binary | | 256, ternary | | | 512, binary | | | 512, ternary | | |
|---|---|---|---|---|---|---|---|---|---|---|---|
| $h$ | 5 | 6 | 4 | 5 | 6 | 57 | 59 | 63 | 55 | 58 | 60 |
| Seq2seq | 7 | 4,7 | 0,5,7,18 | 0,0,1 | 1 | 4 | 8 | 7 | 16 | 11 | - |
| Encoder-only | 20,23,27 | 16,23,28 | 0,16,27 | 1,1,2,6 | 2 | 2 | 3 | 3 | - | 6 | 5 |

Overall, we find that the performance of the two models are comparable (see Table 8). The encoder-only model requires more training epochs before recovery for $n = 256, \log_2 q = 12$, but requires fewer epochs for $n = 512$, and may scale well to larger $n$. Furthermore, we observe that the EMD distinguisher still enables full recovery of ternary secrets with high $h$.

## 8 Related Work

**ML for cryptanalysis.** Numerous proposals leverage ML for cryptanalysis, either indirectly or directly. We call approaches which use ML as part of (but not the main element of) the cryptanalysis process *indirect* approaches. Indirect approaches typically use ML models to strengthen existing cryptanalysis approaches, such as side channel or differential analysis [18]. Most relevant to this work is a recent paper showing the successful use of ML algorithms in side-channel analysis to attack Kyber, one of the NIST standardized PQC proposals [28]. *Direct* ML-based cryptanalysis schemes train models to directly recover cryptographic secrets from plaintext/ciphertext pairs or similar information. Such approaches have been studied against a variety of cryptosystems, including block ciphers [29, 9, 18, 3, 53, 38, 8], hash functions [30], and substitution ciphers [1, 54, 7, 31]. The two ML-based LWE attacks described in §1, SALSA [58] and PICANTE [42], fall under this heading.

**Use of transformers for mathematics.** In recent years, language models have been used to solve math problems beyond the cryptanalysis applications explored in this work. Much prior work has considered how well language models can solve written math problems [35, 48]. More recently, [32] showed large transformers could achieve high accuracy on elementary/high school problems, and [59] explored the accuracy of GPT-3 [11] on math word problems. Language models have also been applied to formalized symbolic math problems. After [40] demonstrated that transformers can solve such problems with high accuracy, follow-up work has explored transformers' use in theorem proving [43], dynamical systems [13], SAT solving [51], transport graphs [14], and symbolic regression [10, 23]. Finally, some have proposed customizing model architectures to enable specific arithmetic operations [36, 56, 44].

# 9    Discussion

We present VERDE, a ML-based attack on LWE with sparse small secrets. VERDE improves data pre-processing and secret recovery, enabling significant performance gains over prior work, SALSA [58] and PICANTE [42]. In particular, VERDE can recover secrets with more general distributions (ternary, small Gaussian), larger dimensions $n$, smaller moduli $q$, and higher $h$. In our implementation, VERDE outperforms the uSVP attack, requiring less time but more compute resources (details in Appendix A.7). Most importantly, this work provides key theoretical insights into observed attack performance, paving the way for targeted future work. Note that even if we recover secrets with seemingly low probability, such as one seed out of ten succeeds for that secret $(1/10)$, or if we do not recover all secrets successfully, such as we recover one out of ten secrets with our attack, this is still enough to make these cryptosystems unsafe to use with low weight secrets (unless there is a way to check for vulnerability to our attack without running the attack).

**Limitations and broader impact.** Despite significantly advancing the state-of-the-art in ML-based LWE attacks, VERDE cannot yet break standardized LWE-based PQC schemes, limiting its real-world impact. Because of this, our paper raises no immediate security concerns. Nevertheless, we have shared a copy of our paper with the NIST PQC group to make them aware of this attack.

**Future work.** Scaling VERDE to attack real-world systems requires work in several directions: increasing dimension $n$, reducing modulus $q$, and increasing $h$, the number of nonzero entries in recoverable secrets. Continued innovations in model architecture (a la §7) may allow attacks in higher dimension $n$, while the theoretical scaling law of §6 provides helpful guidance for improving $q$ and $h$. Given our new insight and analysis of the importance of the **NoMod** percentage, we conclude that the reason that small $q$ is hard for the transformers with our current approach is that they are not good at modular arithmetic (not yet, anyway—this is an area for future work and improvement). In addition, our preprocessing is not well-suited to what is precisely needed and relies on existing lattice reduction algorithms which scale poorly with $n$ and $q$.

This suggests two avenues for improvement: first, to develop model architectures that perform modular arithmetic better. Limited existing work has studied the application of ML to modular arithmetic more broadly [44, 33], Second, alternative preprocessing techniques should be developed to directly concentrate the distribution of random vectors, without relying on lattice reduction methods. With the goal of reducing the standard deviation of the training data, around any center, techniques from the broader math community may prove helpful.

**Acknowledgements.** We thank Jana Sotáková for her contributions to developing the attack on ternary secrets, Hamming reduction techniques, and Section A.3. We also thank Mark Tygert for his helpful input.

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

# A Appendix

## A.1 Parameters

**Table 9. LWE, *preprocessing, and training parameters*.** *For the adaptive increase of preprocessing parameters, we start with blocksize $\beta_1$ and LLL-delta $\delta_{LLL1}$, and upgrade to $\beta_2$ and $\delta_{LLL2}$ at a later stage. Parameters base B and bucket size r are used to tokenize the numbers for transformer training.*

| $n$ | $\log_2 q$ | $q$ | $\beta_1$ | $\delta_{LLL1}$ | $\beta_2$ | $\delta_{LLL2}$ | base $B$ | bucket size $r$ |
|---|---|---|---|---|---|---|---|---|
| | 12 | 3329 | 35 | 0.99 | 40 | 1 | 417 | 1 |
| | 14 | 11197 | 35 | 0.99 | 40 | 1 | 1400 | 1 |
| 256 | 16 | 42899 | 35 | 0.99 | 40 | 1 | 5363 | 4 |
| | 18 | 222553 | 35 | 0.99 | 40 | 0.99 | 27820 | 16 |
| | 20 | 842779 | 35 | 0.99 | 40 | 0.99 | 105348 | 64 |
| 350 | 21 | 1489513 | 30 | 0.96 | 40 | 0.99 | 186190 | 128 |
| | 27 | 94056013 | 30 | 0.96 | 40 | 0.99 | 5878501 | 4096 |
| 512 | 41 | 2199023255531 | 18 | 0.93 | 22 | 0.96 | 137438953471 | 134217728 |

VERDE runs data preprocessing with the parameters shown in Table 9 in parallel using multiple CPUs. We fully parallelize when the time to process one matrix is greater than 24 hours—e.g., for $n = 350, \log_2 q = 27$, we used 5000 CPUs. Otherwise, we parallelize on fewer CPUs depending on the attack time allowed—e.g., preprocessing is completed on 270 CPUs in less than 4 days for $n = 256, \log_2 q = 14$, and on 990 CPUs in less than 3 days for $n = 256, \log_2 q = 18$.

In VERDE, the tokenization used by the transformer mirrors the strategy in §4 of [42], but uses smaller bases $B$ and larger bucket sizes $r$ for better performance (Table 9, 10 of [42]). Decreasing $B$ is further supported by Table 10, evaluated on a low modulus data using VERDE's preprocessing ($n = 256, \log_2 q = 16$). Full ternary secret recovery and partial Gaussian secret recovery both improve with smaller $B = 5363$.

**Table 10. *Secret recovery for different bases*.** $n = 256, \log_2 q = 16$. *We show full ternary secret recovery and partial Gaussian secret recovery when using $B = 7150$ and $5363$ on the same datasets and secrets.*

| | ternary | | | | | Gaussian | | | |
|---|---|---|---|---|---|---|---|---|---|
| $h$ | 8 | 9 | 10 | 11 | 12 | 3 | 4 | 5 | 6 |
| $B = 7150$ | 2/10 | 0/10 | 1/10 | 0/10 | 1/10 | 5/10 | 6/10 | 1/10 | 1/10 |
| $B = 5363$ | 2/10 | 0/10 | 2/10 | 0/10 | 2/10 | 5/10 | 6/10 | 1/10 | 3/10 |

## A.2 More seeds for initialization

The ML attacks benefit from running multiple times with different seeds, or *initializations*, as was demonstrated in PICANTE [42, Section 6.4]. More seeds improve both the success probability and the number of epochs required. Table 11 shows how binary secret recovery improves with more seeds, for different Hamming weights $h$ when $n = 256$ and $\log_2 q = 12$.

**Table 11. *Secret recovery with 1 vs 5 seeds*.** $n = 256, \log_2 q = 12$, *binary secrets. For 1 seed, epoch is the epoch of secret recovery; For 5 seeds (ran on the same secrets as the 1 seed experiments), epoch is the lowest epoch of secret recovery among the 5 initializations for each secret.*

| $h$ | 3 | 4 | 5 | 6 |
|---|---|---|---|---|
| recovery, 1 seed | 5/10 | 1/10 | 1/10 | 1/10 |
| *epoch* | 0,0,0,8,17 | 0 | 17 | 13 |
| recovery, 5 seeds | 7/10 | 3/10 | 1/10 | 2/10 |
| *epoch* | 0,0,0,7,7,8,17 | 0,5,5 | 7 | 4,7 |

Table 12 shows the results for ternary secrets on $n = 512, \log_2 q = 41$, where we run 5 initializations for each secret. While most initializations partially recovered the secret, only a few got full recovery within 20 epochs. Full recovery benefits from more initializations, especially for high $h$.

**Table 12.** *Ternary secret recovery with 5 initializations.* $n = 512$. '-': recovery did not occur in $\leq 20$ epochs.

| $h$ | 45 | 46 | 50 | 55 | 58 | 60 |
|---|---|---|---|---|---|---|
| epoch of partial recovery | 4,4,6,6,7 | 1,1,1,1,1 | 3,4,5,7,- | 5,6,6,6,7 | 6,6,8,10 | 5,8,8,8,12 |
| epoch of full recovery | 10,-,-,-,- | 5,7,-,-,- | 4,10,14,17,- | 16,-,-,-,- | 11,11,-,-,- | -,-,-,-,- |

## A.3 Comparison with PICANTE

To demonstrate the power of the new preprocessing in SALSA VERDE, we run a set of experiments on $n = 256$, $\log_2 q = 23$, where PICANTE also showed success. Using blocksize $\beta = 40$, each matrix is processed by VERDE in about 1.5 days; PICANTE took 2.2 days/matrix. VERDE achieves a reduction factor of 0.25, compared to 0.33 in PICANTE. As shown in Table 1, the difference in the preprocessing step is even more striking for lower $q$.

The highest $h$ recovered by PICANTE was $h = 31$ in 4 out of 20 experiments; VERDE recovered $h = 43$. In Table 13, we see that for $h = 26 - 31$, VERDE significantly outperforms PICANTE in both the success rate and the number of epochs required. In other words, better preprocessing results in lower training time and better secret recovery.

**Table 13. Epochs of secret recovery for PICANTE vs. VERDE.** $n = 256$ and $\log_2 q = 23$. '-' means secret not recovered. 5 secrets per $h$, except for PICANTE $h = 31$ (20 secrets).

| $h$ | 26 | 27 | 28 | 29 | 30 | 31 |
|---|---|---|---|---|---|---|
| PICANTE | 2,3,4,7,- | 10,-,-,-,- | 5,-,-,-,- | 5,9,11,-,- | 17,20,32,-,- | 6,12,26,27 (out of 20 secrets) |
| VERDE | 0,0,2,2,7 | 2,6,-,-,- | 0,0,0,1,2 | 0,1,1,1,- | 0,1,2,3,- | 1,2,3,4,- |

## A.4 Distinguisher tested on reduced data outperforms random data

We compare the performance of running the distinguisher on the preprocessed data that were held out from the training set ($\text{Dist}_{\text{BKZ}}$) with running on random vectors ($\text{Dist}_{\text{rand}}$). We run both set of experiments with the same model initialization seeds, and record a success for the method(s) that recovers the secret at the earliest epoch. Table 14 indicates that $\text{Dist}_{\text{BKZ}}$ performs better.

**Table 14.** *Secret recovery by running the distinguisher on the random vectors and bkz preprocessed data.* $n = 256$, $\log_2 q = 23$, *on data processed using* PICANTE*'s approach.*

| $h$ | 27 | 28 | 29 | 30 | 31 |
|---|---|---|---|---|---|
| $\text{Dist}_{\text{BKZ}}$ | 2/5 | 1/5 | 1/5 | 0/5 | 0/5 |
| $\text{Dist}_{\text{rand}}$ | 1/5 | 0/5 | 0/5 | 0/5 | 0/5 |

## A.5 Dimension reduction techniques

Most of the entries in a sparse secret are zero, so prior work [5] has suggested the idea of randomly assuming a subset of the entries to be zero and removing them to reduce the dimension of the lattice problem. The assumption will be correct with some probability depending on the secret's sparsity. Here we explore an improvement on this strategy: we use the partially trained model to glean signal on which entries should be kicked out. We can either try to kick out zeros, which we call *dimension reduction*, or in the binary case, kick out 1s, which we call *Hamming reduction*, or combined.

This technique will be better than random when the model has begun to learn information about the bits, reflected in their relative rankings. Specifically, the ranking strategies described in [42, Section 4.3] are used to compute scores which estimate the likelihood of secret bits being 1. Once the model has started to learn, we can assume that the highest ranked bits will correspond to secret bits which are equal to 1, and the lowest ranked bits will correspond to zeros. So we use this information to reduce the dimension of the problem by kicking out low-ranked bits which we guess to be zero or high-ranked bits which we guess to be 1. Then, we retrain a model on the smaller dimensional samples and hope to recover the secret. If the original kicked out bits were correct and the model recovers the secret of the smaller dimensional problem, then we find the original secret.

**Dimension reduction.** Since there are many more 0s than 1s in sparse secrets, we can potentially reduce the dimension significantly. Once we remove the bits with low scores, we can simply re-run

training on the dataset with $(\mathbf{a}', b)$ where $\mathbf{a}'$ are the samples with the corresponding bits removed. If the indices have been identified incorrectly, then the reduction will fail. For $n = 256, \log_2 q = 14$, VERDE attempted 10 binary secrets with $h = 10$ and did not recover the secret. Then we tried dimension reduction on these experiments and recovered one secret.

**Hamming reduction.** Kicking out $1s$ from the secret is particularly valuable, given the theoretical analysis of the VERDE in Section 6. If nonzero bits are indeed ranked at the top by the model, a straightforward approach of kicking out the top-ranked bits and retraining on the smaller dimension and Hamming weight will likely yield improved secret recovery.

But in case some of the top-ranked bits are not equal to 1, we propose the following strategy. Suppose $S$ is a small set of indices for bits with the highest scores. We construct the following problem: let $\mathbf{s}'$ be $\mathbf{s}$ with bits in $S$ flipped, and $\mathbf{a}'$ be $\mathbf{a}$ with $a_i$ negated for $i \in S$. Equivalently, for $i \in S$, $a_i' = -a_i$ and $s_i' = 1 - s_i$. Then, the corresponding

$$b' = \mathbf{a}' \cdot \mathbf{s}' = \sum_{i \notin S} a_i s_i + \sum_{i \in S} a_i' s_i' = \sum_{i \notin S} a_i s_i + \sum_{i \in S} -a_i(1 - s_i) = b - \sum_{i \in S} a_i.$$

If more than half of the indices in $S$ are 1, then $\mathbf{s}'$ has a smaller Hamming weight, hence the instance $(\mathbf{a}', b' = b - \sum_{i \in S} a_i)$ is likely easier. If exactly half of the indices in $S$ are 1, then $\mathbf{s}'$ has the same $h$ as $\mathbf{s}$, but the new instance $(\mathbf{a}', b')$ will have a different **NoMod** and may be recoverable.

## A.6 Attacking sparse secrets in larger dimensions

For sparse secrets on even larger dimensions, we can apply our attack after using combinatorial techniques to exploit the sparsity of the secret. The approach would be to combine VERDE with the techniques from [5, 20] as follows: Randomly kick out $k$ entries of the secret, assuming they are zero, which will be true with some probability. This reduces the LWE problem to a smaller dimension where VERDE can recover the secret. The expected cost of the attack would be VERDE's cost multiplied by $1/p$, where $p = (\frac{n-h}{n})^k$ is the probability that the assumption that the $k$ entries are 0 is correct.

## A.7 Comparison with lattice reduction/uSVP attacks

In this section we compare VERDE with classical lattice reduction attacks in two ways. The LWE Estimator [6] gives heuristic predicted running times for the best-known lattice reduction attacks. But even the authors of the LWE Estimator claim that the estimates are often wrong and unreliable. So we compare VERDE to the Estimator results but we also compare to concrete running times in the case $n = 256$, achieved by implementing the uSVP lattice attacks ourselves, on the same machines where we run our ML-based attacks. Unfortunately, many attacks listed by the LWE Estimator lack practical/accessible implementations, and we lacked the time and resources to implement all of these and run comparisons. Thus, we focus our concrete comparisons on the uSVP attack, which the Estimator predicts to be the best method for larger $n$ such as $n = 512$. We leave comparison of VERDE against other lattice reduction attacks as important future work for the broader lattice community, as practical attack run-times are poorly understood, especially for small sparse secrets.

**Table 15. Concrete comparison of VERDE and uSVP attacks for $n = 256$, binary secrets, varying $q$ and $h$.** VERDE's total attack time is the sum of preprocessing and training time (with recovery included). Preprocessing time assumes full parallelization, and training time is the number of epochs to recovery multiplied by epoch time (1.5 hours/epoch). **fail** means no successful secret recovery for uSVP to compare to.

| LWE parameters | | VERDE attack time | | | uSVP attack time (hrs) |
|---|---|---|---|---|---|
| $\log_2 q$ | $h$ | Preprocessing (hrs) | Training | Total (hrs) | |
| 12 | 8 | 1.5 | 2 epochs | 4.5 | **fail** |
| 14 | 12 | 2.5 | 2-5 epochs | 5.5-10 | **fail** |
| 16 | 14 | 8.0 | 2 epochs | 11 | **fail** |
| 18 | 18 | 7.0 | 3 epochs | 11.5 | 558 |
| 18 | 20 | 7.0 | 1-8 epochs | 8.5-19 | 259 |
| 20 | 22 | 7.5 | 5 epochs | 15 | 135-459 |
| 20 | 23 | 7.5 | 3-4 epochs | 12-15 | 167-330 |
| 20 | 24 | 7.5 | 4 epochs | 13.5 | 567 |
| 20 | 25 | 7.5 | 5 epochs | 15 | 76 - 401 |

Table 15 presents our concrete comparison between VERDE and the uSVP attack on binary secrets for $n = 256$. To summarize the comparison, VERDE outperforms the uSVP attack in two senses: 1) VERDE fully recovers sparse binary and ternary secrets for $n$ and $q$ in some cases where the uSVP attack does not succeed in several weeks or months using *fplll* BKZ 2.0 [19] with the required block size; and 2) in cases where we improve the uSVP implementation enough (see below) to run with the required large block size, we find that in all cases, VERDE recovers the secrets much faster.

**Summarizing classical lattice reduction attacks.** Table 16 gives the estimated heuristic cost and specifies the block size for attacking sparse binary and ternary secrets for $n = 256, 350, 512$ and various $q$ with the best known classical lattice reduction attack.

**Table 16.** *Estimated cost of best classical attack (*LWE *Estimator). For* VERDE*'s highest $h$, we run the* LWE *Estimator and report the estimated cost and block size $\beta$ for the best predicted attack.*

| $n$ | $\log_2 q$ | binary secret | | | | ternary secret | | | |
|---|---|---|---|---|---|---|---|---|---|
| | | h | best attack | rop | $\beta$ | h | best attack | rop | $\beta$ |
| 256 | 12 | 8 | dual_mitm_hybrid | $2^{43.0}$ | 40 | 9 | dual_mitm_hybrid | $2^{43.8}$ | 40 |
| | 14 | 12 | dual_hybrid | $2^{47.2}$ | 40 | 13 | dual_hybrid | $2^{47.7}$ | 41 |
| | 16 | 14 | dual_hybrid | $2^{46.8}$ | 40 | 16 | dual_hybrid | $2^{47.4}$ | 41 |
| | 18 | 23 | bdd_hybrid | $2^{47.1}$ | 45 | 23 | dual_hybrid | $2^{47.4}$ | 41 |
| | 20 | 36 | bdd | $2^{44.0}$ | 45 | 33 | bdd | $2^{44.2}$ | 46 |
| 350 | 21 | 12 | dual_mitm_hybrid, bdd_mitm_hybrid | $2^{46.4}$ | 40 | 13 | dual_mitm_hybrid | $2^{46.9}$ | 54 |
| | 27 | 36 | bdd | $2^{44.9}$ | 47 | 38 | bdd, bdd_hybrid | $2^{44.1}$ | 43 |
| 512 | 41 | 63 | usvp/bdd | $2^{42.9}$ | 40 | 60 | usvp/bdd | $2^{42.9}$ | 40 |

**Table 17.** *Estimated cost and block sizes of uSVP. $n = 256, \log_2 q = 16, 18, 20$, for binary, ternary, and Gaussian secrets with $h$ nonzero entries.*

| $\log_2 q$ | binary | | | ternary | | | Gaussian | | |
|---|---|---|---|---|---|---|---|---|---|
| | h | rop | $\beta$ | h | rop | $\beta$ | h | rop | $\beta$ |
| 16 | 12 | $2^{54.7}$ | 86 | 12 | $2^{54.9}$ | 87 | 6 | $2^{69.3}$ | 138 |
| 18 | 18 | $2^{49.5}$ | 67 | 19 | $2^{49.8}$ | 68 | 7 | $2^{59.5}$ | 102 |
| 20 | 25 | $2^{45.4}$ | 52 | 24 | $2^{45.4}$ | 52 | 7 | $2^{53.5}$ | 80 |

Table 17 gives the same information for the uSVP classical lattice reduction attack, focusing on $n = 256$ for some of the larger $q$ and $h$ where VERDE succeeds.

**uSVP attack performance, binary secrets.** For $n = 256$ and $\log_2 q = 20$, we run the concrete uSVP attack using *fplll* BKZ 2.0 with Kannan's embedding and parameters ([37], [16]). With block size 50 and 55, secrets are not recovered in 25 days, and block size 60 or larger cannot finish the first BKZ loop in 3 days. We propose two improvements to get these attacks to run faster and better: 1) we rearrange the rows of the uSVP matrix as in VERDE; 2) we use the adaptive float type upgrade as in VERDE. In addition, after each BKZ loop, we also run the secret validation obtained from the shortest vector found so far, and terminate if we get a secret match.

With rearranging the rows but without the adaptive float type upgrade, we have to use higher precision because otherwise the attack fails due to low precision after running for 23 hours. The attacks run quite slowly with high precision and did not recover secrets after 25 days, except in one case where a binary secret with $h = 22$ was recovered with block size 55 in 414 hours, roughly 17 days.

With rearranging the rows and the adaptive float type upgrade, we ran $n = 256$ and $\log_2 q = 20$ with block size $50 - 55$ for binary/ternary secrets, and $n = 256$ and $\log_2 q = 18$ with block size 65 for binary/ternary secrets. See Table 18 for running times and $h$. For example, for $\log_2 q = 20$, a ternary secret with $h = 25$ was recovered in 280 hours, roughly 12 days, and for $\log_2 q = 18$, a binary secret with $h = 20$ was found in $\approx 11$ days. When the block size used is lower than predicted by the estimator (50 instead of 52 for $\log_2 q = 20$ and 65 instead of 67 for $\log_2 q = 18$, see Table 17, 18), the uSVP attack succeeds in recovering only a few secrets out of many which were tried.

**Table 18.** *uSVP concrete attack time,* $n = 256$. *Upper:* $\log_2 q = 20$, *lower:* $\log_2 q = 18$. *For each $h$, block size and secret distribution, we run 5 experiments on different secrets and show the secret recovery time (in hours).* '-' *means no success in 25 days.*

| | | $n = \mathbf{256}, \log_2 \mathbf{q} = \mathbf{20}$ | | | |
|---|---|---|---|---|---|
| blocksize | secret | $h = 22$ | $h = 23$ | $h = 24$ | $h = 25$ |
| 50 | binary | - | - | - | 451, 531 |
| | ternary | 261, 367 | - | - | - |
| 55 | binary | 135, 161, 459 | 167, 323, 330 | 567 | 76, 280, 401 |
| | ternary | 43, 241, 432 | 234, 296 | 226, 257, 337 | 280 |

| | | $n = \mathbf{256}, \log_2 \mathbf{q} = \mathbf{18}$ | | | | | |
|---|---|---|---|---|---|---|---|
| blocksize | secret | $h = 11$ | $h = 12$ | $h = 17$ | $h = 18$ | $h = 19$ | $h = 20$ |
| 65 | binary | 32 | - | - | 558 | - | 259 |
| | ternary | 324, 553 | 109 | 594 | - | - | - |

The concrete experiments allow us to validate to some extent the predictions of the Estimator in many cases, giving confidence in the comparison with VERDE. Running the uSVP attack with block size 70 or larger didn't finish the first BKZ loop in 3 days, so we expect longer attack time for Gaussian secrets and for binary and ternary secrets with lower $q$.

**Gaussian secrets.** VERDE achieves partial Gaussian secret recovery for small $h$, reducing the secret recovery to a lattice problem in tiny dimension ($h$). Because the preprocessing time and per epoch time does not vary with secret distribution, VERDE's attack time on Gaussian secrets is comparable to on binary secrets (see Table 15, 19). We note that preprocessing takes longer for $n = 350$ and $\log_2 q = 27$ due to precision issues. This is an important observation because lattice problems are supposed to be harder for smaller $q$. In contrast, with classical attacks, the Estimator predicts significantly larger block sizes required, and longer running times (Table 20), than for binary secrets. VERDE compares favorably to classical attacks on Gaussian secrets, especially on small $q$ (where the problem is harder), e.g., VERDE recovers Gaussian secrets with $h = 5 - 6$ in $4.5 - 15$ hours for $n = 256, \log_2 q = 12$, where the cost of best classical attacks is predicted to be $2^{91}$ rop.

**Table 19.** VERDE's *performance on* LWE *problems with* $n = 256$ *and* 350, **Gaussian secrets,** *varying $q$ and $h$. Preprocessing time: hours to process one matrix. Total attack time: sum of preprocessing time (assuming full parallelization) and training time (number of epochs multiplied by hours per epoch, see §2).*

| **LWE parameters** | | | **VERDE attack time** | | | |
|---|---|---|---|---|---|---|
| $n$ | $\log_2 q$ | $h$ | Preprocessing (hrs) | Training | hrs/epoch | Total (hrs) |
| | 12 | 5-6 | 1.5 | 2-9 epochs | | 4.5-15 |
| | 14 | 5-6 | 2.5 | 2-21 epochs | | 5.5-34 |
| 256 | 16 | 9 | 8.0 | 2 epochs | 1.5 | 11 |
| | 18 | 9 | 7.0 | 3 epochs | | 11.5 |
| | 20 | 10 | 7.5 | 5 epochs | | 15 |
| 350 | 21 | 5 | 16 | 1-5 epochs | 1.6 | 18-24 |
| | 27 | 10 | 216 | 2-13 epochs | | 219-237 |

**Table 20.** LWE *Estimator: best classical attack on Gaussian secrets on various* $q$. $n = 256$ *and* 350. *For* VERDE's *highest $h$, we show the best classical attack, the attack cost (rop), and predicted block size $\beta$ from the* LWE *Estimator.*

| $n$ | $\log_2 q$ | $h$ | best attack | rop | $\beta$ |
|---|---|---|---|---|---|
| | 12 | 6 | bdd / bdd_hybrid | $2^{91.0}$ | 214 |
| | 14 | 6 | bdd / bdd_hybrid | $2^{77.7}$ | 166 |
| 256 | 16 | 9 | bdd / bdd_hybrid | $2^{67.1}$ | 128 |
| | 18 | 9 | bdd | $2^{57.7}$ | 93 |
| | 20 | 10 | bdd / bdd_hybrid | $2^{51.9}$ | 72 |
| 350 | 21 | 5 | bdd | $2^{65.2}$ | 120 |
| | 27 | 10 | bdd / bdd_hybrid | $2^{50.2}$ | 68 |

