# OpenReview forum: "SALSA VERDE: a machine learning attack on LWE with sparse small secrets"
_NeurIPS.cc/2023/Conference — NeurIPS 2023 poster_

### Official Review · Reviewer_heZ1 · 2023-06-10

**Soundness:** 3 good
**Presentation:** 3 good
**Contribution:** 3 good
**Rating:** 6
**Confidence:** 2

**Summary:**

This paper introduces a novel ML attack that has the capability to recover sparse binary, ternary, and small Gaussian secrets. The proposed approach exhibits effectiveness in attacking LWE systems with larger dimensions and smaller moduli, while requiring less preprocessing time.

**Strengths:**

1. The paper is well-written.
2. The SALSA VERDE achieves a harder attack with larger dimensions and smaller moduli, which makes the attack more practical on LWE-based systems.
4. The improved preprocessing method significantly speeds up the attack process, improving its efficiency.
5. The authors provide a theoretical explanation for the performance of the attack, adding to its overall understanding.

**Weaknesses:**

1. It would be beneficial to include an explanation or insights regarding the reasons why full recovery of Gaussian secrets is infeasible.

2. More ablation experiments and comparisons should be conducted to showcase the individual contributions and benefits of each proposed enhancement.

3. Regarding the model architecture, additional results or explanations are necessary to justify the replacement of the positional embedding method and beam search, especially when the performances are similar.

**Questions:**

Please refer to the weaknesses section.

**Limitations:**

Please refer to the weaknesses section.

---

> ### Author Rebuttal · Authors · 2023-08-08
>
> **It would be beneficial to include an explanation or insights regarding the reasons why full recovery of Gaussian secrets is infeasible.**
>
> Full recovery of Gaussian secrets is feasible. Point 2 of our general rebuttal describes three possible approaches. We will include these in the revised version of the paper.
>
> **More ablation experiments and comparisons should be conducted to showcase the individual contributions and benefits of each proposed enhancement.**
>
> Such results already exist, but they are scattered throughout the paper. In particular, the impact of new preprocessing techniques is presented in table 2. The impact of improved preprocessing reduction on secret recovery is presented in table 5. The impact of the new distinguisher is in Table 14 of the appendix,  the effect of the ternary distinguisher is discussed in section 3, and that of the encoder only model in table 8. We will streamline these results, and add a synthesis in the discussion of the revised paper.
>
> **Regarding the model architecture, additional results or explanations are necessary to justify the replacement of the positional embedding method and beam search, especially when the performances are similar.**
>
> We agree that additional work is needed on the encoder-only approach. We found it interesting to report that such architectures can succeed: so far, all previous papers used seq2seq transformers.

---

### Official Review · Reviewer_j97t · 2023-07-03

**Soundness:** 3 good
**Presentation:** 2 fair
**Contribution:** 3 good
**Rating:** 7
**Confidence:** 3

**Summary:**

This paper considers the problem of adapting machine learning for the cryptanalysis of LWE --- an important cryptographically hard problem that forms the basis of several modern post-quantum cryptosystems. The LWE problem requires recovery of a secret \bf{s} given many noisy inner products b = \bf{a} \cdot \bf{s} + e mod q.  LWE is also an ideal candidate problem for cryptanalysis using ML techniques because of the linear structure in the computation.

The high-level approach is to train an ML model using samples {(\bf{a}, b)}. Next, this trained model can predict a value close to \bf{a} \cdot \bf{s}  mod q for new values of \bf{a}.  Finally, this prediction can recover \bf{s}. This approach was first considered in prior work Salsa/Picante. Salsa could only work for binary secrets with 3/4 non-zero entries. In contrast, Picante supplemented the above approach with some insights from lattice algorithms. Specifically, Picante showed that by appropriately using BKZ lattice reduction algorithm in preprocessing, they could attack binary secrets but with larger non-zero entries.

However, a significant gap remains between the parameter choices these attacks can handle and those used in LWE-based cryptosystems. This paper makes progress in reducing this gap.

**Strengths:**

+ The paper makes progress on various strengths.
+ The paper can beat state-of-the-art lattice attacks not using ML. This demonstrated that ML could play a crucial role in the pipeline for improvement on lattice attacks.
+ I like the effort in trying to give a theoretical explanation of why the attacks only seem to work for working

**Weaknesses:**

- The paper is a bit dense and talks about various parameters. Having some text that helps provide a unifying framework from the various parameter choices the authors try to use will be helpful.


**Questions:**

- How do you quantify that your attacks are better than classical lattice reduction attacks? I missed the concrete performance comparison on the same. Could the performance boost be attributable to more optimizations/use of hardware specialized for ML tasks?
- Several choices in the paper seem unexplained. For example, the authors on page 5 say that they use \sigma = 3 and that FHE standards suggest \sigma = 3.2. I understand the authors are using a \sigma close to what is suggested, but why not use 3.2?

**Limitations:**

- The improvements in the paper are still not significant enough to show that ML/transformers can help significantly improve the attacks against LWE.

---

> ### Author Rebuttal · Authors · 2023-08-08
>
> **The paper is a bit dense and talks about various parameters. Having some text that helps provide a unifying framework from the various parameter choices the authors try to use will be helpful.**
>
> We agree. Section A.1 in the appendix provides some background on the LWE instances and preprocessing parameters, but we will complete it with the training parameters, have a more detailed discussion of parameter choices and add it to the main.
>
> **How do you quantify that your attacks are better than classical lattice reduction attacks?**
>
> The comparisons are in the appendix, notably sections A.3 and A.7.
>
> Table 15 shows that, for dimension 256, when run on the same machines, and using the same lattice reduction library, our techniques, once parallelized, run faster than classical uSVP attacks. For larger dimensions, we were not able to successfully run uSVP attacks (e.g. they never terminated), whereas we could recover secrets using our techniques. Estimates of the time needed for classical reduction attacks, for n=350 and 512, are in table 16 and 18. Finally we provide comparisons for gaussian secrets in tables 19 and 20.
>
> Finally, for dimensions up to 350,  some comparisons with classical attacks were included in PICANTE, and we prove (in section A.3 of the appendix) that we are better than PICANTE.
>
> **Several choices in the paper seem unexplained. For example, the authors on page 5 say that they use $\sigma = 3$ and that FHE standards suggest $\sigma = 3.2$. I understand the authors are using a $\sigma$ close to what is suggested, but why not use 3.2?**
>
> $\sigma=3$ was used in SALSA and PICANTE, which are the main baselines we use for comparison. We agree that a choice of $3.2$ would have been preferable. Still, we believe that the small difference in error variance has little impact on our results (especially since the preprocessing step amplifies error).
>
> **The improvements in the paper are still not significant enough to show that ML/transformers can help significantly improve the attacks against LWE.**
>
> See point 1 of our general rebuttal. We believe that VERDE establishes a new state of the art on medium to hard LWE instances with sparse binary and ternary secrets (which are considered for standardization). Also, table 15 in appendix A.7 indicates that we outperform classical uSVP attacks.

---

### Official Review · Reviewer_XtRr · 2023-07-15

**Soundness:** 3 good
**Presentation:** 3 good
**Contribution:** 2 fair
**Rating:** 5
**Confidence:** 2

**Summary:**

The authors propose an ML based attack scheme for the learning with errors post quantum cryptography scheme. Their proposed attack is able to recover sparse, binary, and ternary Gaussian secrets in a more efficient fashion than the Salsa and Picante schemes in prior work.

**Strengths:**

I have to admit that I am not an expert in post quantum cryptography, nor I am familiar with previous literature. Therefore, I am not able to confidently appreciate the strengths of the paper with respect to prior work. I like the clarity of the presentation in most parts; the paper is written well and provides a decent introduction to the problem of learning with errors.

**Weaknesses:**

1. To me the purpose of the paper is not clear. The authors claim that they improve on previous schemes. But these schemes have already demonstrated that LWE is breakable and won't be used in practice anyway, so what is the novelty of the paper? This should be made clear in a better way.

2. It seems that the difference to previous schemes is just a different matrix factorization in combination with techniques from [14] and [17], as explained in Section 2. I don't think that this contribution alone is sufficient for acceptance to NeuRIPS.

3. The part of the paper related to learning is not well developed and presented as an afterthought, despite the choice of the architecture may significantly affect the performance. The proposed scheme leverages the same architecture as in previous works without giving any rationale why a transformer architecture is the best choice and why attention is necessarily required. It seems that in the LWE encoding the noise sequence is drawn iid from a Gaussian distribution, so I don't understand why the learning architecture needs to impose a Markovian structure.

**Questions:**

Please see my questions under Weaknesses.

**Limitations:**

The authors have stated the need for post quantum cryptography, and thus, the societal impact of breaking classical cryptography via Shor's algorithm. Their work contributes to the task of finding the best post quantum crypto scheme which does not depend on factoring large numbers into products of primes.

---

> ### Author Rebuttal · Authors · 2023-08-08
>
> **To me the purpose of the paper is not clear. The authors claim that they improve on previous schemes. But these schemes have already demonstrated that LWE is breakable and won't be used in practice anyway, so what is the novelty of the paper?**
>
> The general context of this paper is the current standardization of LWE-based cryptosystems, as possible replacements for public key algorithms such as RSA which are vulnerable to Shor’s algorithm. In its strongest form (non-sparse secrets with coordinates uniformly sampled in [0,q)), LWE is not breakable. However, sparse secrets have been considered because they make computation faster, and binary secrets are desirable because they enable homomorphic encryption (HE): performing computations on encrypted data. As of today, one of the schemes proposed for HE involves sparse ternary secrets with dimension 1024 or more. Previous work (PICANTE) suggested that ML-based attacks can succeed for dimension up to 350 and binary secrets. The purpose of this paper is to scale these attacks, to recover larger dimensions (512), lower values of q, and ternary secrets. As such, it establishes a new state of the art on medium-to-hard LWE problems, with sparse binary or ternary secrets.
>
> See also part 1 of our general rebuttal. We will clarify this in the introduction of the revised paper.
>
> **It seems that the difference to previous schemes is just a different matrix factorization in combination with techniques from [14] and [17], as explained in Section 2.**
>
> Our introduction lists 6 contributions. Together with the acceleration of preprocessing described in section 4, we make the other important contributions:
> * The two bit distinguisher discussed in section 3, which allows to recover new classes of sparse secrets: ternary, and small secrets (when extended with the techniques described in our reply to reviewers X and Y). We demonstrate that ternary secrets are as easy to recover as binary.
> * We provide a theoretical analysis of these techniques, in section 5 and 6. First, we explain (via the NoMod statistics) why some secrets are harder to recover than others. Then we discuss the impact of preprocessing on recovery, and how it scales with the Hamming weight.
> * We scale previous attacks to larger dimensions (512) and lower modulus (lg q=12), establishing a new SOTA (see discussion in section A.7)
> * We demonstrate that seq2seq architectures can be replaced by encoder-only models, without adversely affecting the performance of the attack.
>
> **The part of the paper related to learning is not well developed and presented as an afterthought, despite the choice of the architecture may significantly affect the performance. The proposed scheme leverages the same architecture as in previous works without giving any rationale why a transformer architecture is the best choice and why attention is necessarily required. It seems that in the LWE encoding the noise sequence is drawn iid from a Gaussian distribution, so I don't understand why the learning architecture needs to impose a Markovian structure.**
>
> We believe that an attention-based encoder is necessary, because predicting $b=\textbf a \cdot •textbf s+e$ (with **s** an unknown vector) from **a**, a random vector, requires understanding how the different coordinates of **a** interact during the calculation of b. The authors of SALSA also mention the failure of MLP and LSTM. On the other hand, we agree that the auto-regressive decoder introduced in SALSA is not necessary, because the choice of a base larger than the square root of $q$ means that the model output only has two tokens.
>
> Decoder models such as GPT 3.5/4 are known to have difficulties to generate direct answers to certain --- sometimes even simple --- problems without generating chain of thoughts first. Our situation is analogous, since we are generating just one or two output tokens, so we want the transformer to have full thinking power before outputting the answer. Letting the transformer be encoder-only can be more directly tackling our LWE problem, by reducing the unnecessary components, and letting the transformer, in all layers, have more incentive to let all pairs of input tokens interact.This was the main driver for our investigation of encoder-only models.
>
> An additional benefit of an encoder-only model is that it uses less parameters, which might help scaling to larger dimensions (and longer input sequences). We will clarify our reasons for introducing the encoder-only architecture.

---

### Official Review · Reviewer_StdV · 2023-07-23

**Soundness:** 4 excellent
**Presentation:** 4 excellent
**Contribution:** 3 good
**Rating:** 7
**Confidence:** 3

**Summary:**

The authors present VERDE, an improved ML based attack on the LWE problem, a source of mathematical hardness underpinning post-quantum cryptography schemes such as Homomorphic Encryption. Their method improves upon prior versions of this form of attack, SALSA and PICANTE, by improving the runtime characteristics of the preprocessing and the performance and generality of the secret recovery step. Additionally, their error analysis yields insights into the hardness of SALSA-style attacks on the LWE problem, revealing an avenue for potential improvement based on improving the distribution of the training data to increase the chance of success for such attacks.

**Strengths:**

- Introduces key performance improvements to the existing previous art by subtle but impactful modifications: reducing the complexity of the preprocessing matrix operations, and improving their floating point error tolerance.

- Secret discovery procedure is improved by applying a simple (but not trivial) fact from machine learning - generalization to a shifted distribution is hard. Switching to verification on a held out set of preprocessed vectors. i.e. "in-distribution", yields improved performance.

- Secret discovery procedure is extended to ternary secrets by taking a novel two step approach, where step one is similar to the binary secret recovery problem. (novelty is taken for granted, not very familiar with literature)

- (Highlight) Error analysis yields NoMod, which allows them to first descriptively develop an explanation for different success rates as a function of modulus, but then further admits a potential improvement via attempting to increase the NoMod factor of a given training set by permutation.

**Weaknesses:**

- The encoder-only model architecture proposal empiricals are mixed, and theoretical or intuitive justification is missing. Would be useful to include training runtime differences in Table 8 or their own table, my sense is that epochs until success doesn't tell full story since training complexity for the two architectures is likely different. Claim that the encoder only model may generalize to larger n is not sufficiently supported by Table 8, could be omitted, or such discussions moved to future work section as more experimentation is necessary.

- (Minor) Rationale for omission of other secret recovery techniques - direct, cross-attention - is not fully convincing. Latter could be included for the seq2seq architecture as they are proposing the encoder-only variant in this work, rather than adopting it as a baseline.

**Questions:**

- Can slightly more detail be provided about full recovery in the Gaussian case? "This is feasible via other methods" leaves reader wondering whether identifying non-zero entities is in fact most of overall lift. A more detailed explanation for how to complete recovery in the gaussian case should be included, else claims that VERDE covers the gaussian case should be qualified elsewhere.

- To strengthen the already decisive insight of the NoMod analysis, can anything be said analytically about the likelihood of a given permutation achieving a high NoMod percentage given the observation of failure on the initial un-permuted version? Working this analysis might reveal an avenue for further tuning of the permutation procedure to increase the success of this trick. Alternately, this could provide an avenue for robustness analysis of this attack under certain assumptions about the distributions of training data.

**Limitations:**

- Key limitation, as the author's state, is gap between the settings they can attack successfully, and the realistic parameter schemes in standardized PQC. However, this is not actually any failure of the authors, and they do in fact achieve results at more realistic settings than prior work.

---

> ### Author Rebuttal · Authors · 2023-08-08
>
> **The encoder-only model architecture proposal empiricals are mixed, and theoretical or intuitive justification is missing.**
>
> The intuition for moving to encoder-only architectures is the fact that model output is only two-token long. This suggests that an autoregressive decoder is not needed, and simpler architectures are possible.
> Also, decoder models such as GPT 3.5/4 are known to have difficulties to generate direct answers to certain --- sometimes even simple --- problems without generating chain of thoughts first. Our situation is analogous, since we are generating just one or two output tokens, so we want the transformer to have full thinking power before outputting the answer. Letting the transformer be encoder-only can be more directly tackling our LWE problem, by reducing the unnecessary components, and letting the transformer, in all layers, have more incentive to let all pairs of input tokens interact.
> An additional potential benefit is the fact that encoder-only models have less parameters, and should be easier to train as dimension scales up.
>
> We will consider moving the encoder-only discussion to the appendix, and reinforcing the sections on full secret recovery, and comparisons with classical attacks.
>
> **Rationale for omission of other secret recovery techniques - direct, cross-attention - is not fully convincing.**
> .
> The main reason for abandoning direct secret recovery was that it almost never recovers secrets in large dimensions when preprocessing is used. We believe this is due to the requirement that the transformer correctly predicts b for values of a with only one non-zero coordinate. As dimension and modulus increase, these points get further from the training distribution, which includes no points with many small coordinates. Direct secret recovery therefore implies OOD generalization. The modified distinguisher used in VERDE, on the other hand, only assumes correct prediction around the training samples, i.e. in-domain generalization, a weaker requirement.
>
> On average, we found that cross-attention performs as well as the modified distinguisher. We removed it to make the encoder-only architecture possible. We will clarify these points in the revised edition.
>
> **Can slightly more detail be provided about full recovery in the Gaussian case?**
>
> Yes. See our general rebuttal, point 2
>
> **To strengthen the already decisive insight of the NoMod analysis, can anything be said analytically about the likelihood of a given permutation achieving a high NoMod percentage given the observation of failure on the initial un-permuted version? Working this analysis might reveal an avenue for further tuning of the permutation procedure to increase the success of this trick. Alternately, this could provide an avenue for robustness analysis of this attack under certain assumptions about the distributions of training data.**
>
> Thank you for this insightful suggestion. Intuitively, a high NoMod happens when the ones in the secret correspond to the coordinates of **a** that were most reduced during preprocessing. We can compute column reduction from the preprocessed sample, and therefore classify the columns in two groups: “easy” (heavily reduced) and “hard” (less reduced). The NoMod analysis suggests that unrecoverable secrets have many ones in the hard columns. A good permutation would therefore exchange the hard columns with easy ones, in the hope that they contain less one bits after reductions. This could help reduce the number of permutations that need to be considered to recover the secret (an important point, since a new BKZ reduction  must be performed for each permutation considered). Note also that this supposes that we can predict (before or early during reduction)  which columns will be reduced. This is a matter for future work.
>
> **Key limitation, as the author's state, is gap between the settings they can attack successfully, and the realistic parameter schemes in standardized PQC. However, this is not actually any failure of the authors, and they do in fact achieve results at more realistic settings than prior work.**
>
> See out general rebuttal, point 1

---

> > ### Comment · Reviewer_StdV · 2023-08-16
> >
> > I have read the rebuttal by the authors and appreciate their thorough responses to my questions.
> >
> > For the draft update, comments regarding intuitions for encoder only models over decoder models should be limited to simple statements based on the concrete requirements of the prediction problem. I would limit the statements to 1) the problem is constrained to two output tokens, so a model capable of generating arbitrary sequences is not required and 2) nothing about the problem setting requires causally masked attention, and instead there is a an argument that a full/bidirectional attention computation is appropriate.
> >
> > I think that the details about the full recovery for gaussian secrets will round out the paper well. I will leave my recommendation as is, at a 7, Accept.

---

### Official Review · Reviewer_wmYQ · 2023-07-25

**Soundness:** 3 good
**Presentation:** 4 excellent
**Contribution:** 3 good
**Rating:** 4
**Confidence:** 4

**Summary:**

This paper continues a recent line of work on leveraging machine learning (specifically, transformers) for cryptanalysis. The focus is specifically on the learning with errors (LWE) problem. The LWE problem is one of the core problems in lattice-based cryptography, and the basis for many of the recently standardized post-quantum cryptographic systems. The paper targets one particular version of LWE (with *sparse* secret keys).

The (search) LWE problem is to learn a noisy linear function, where the noise is over a ring: given samples (A, s^T A + e mod q) for a secret vector s, random (wide) matrix A, and small (discrete) Gaussian noise e, recover the secret s. The variant of LWE considered here is where the secret s is sparse. While this is not the standard version of LWE, it is used as an optimization for some fully homomorphic encryption (FHE) schemes.

Compared to previous approaches (Salsa and Picante), this paper describes generalizations to a ternary and (sparse) Gaussian error distributions as well as a better preprocessing step (responsible for generating more examples via the given LWE challenge, which is in turn used to train the transformer). The paper provides some empirical measurements showing that the attack is successful in recovering the secret for some small instances.

**Strengths:**

Studying and understanding the hardness of lattice assumptions is an important problem in cryptography, especially given the current move towards standardizing and implementing post-quantum cryptography. This paper improves upon an ML-based approach for understanding the hardness of certain LWE instances.

While previous approaches considered the more restrictive setting of LWE with sparse binary secrets, this paper extends the analysis to support LWE instances with sparse ternary secrets. This is a more useful setting as this is a more common mode used in certain FHE implementations (though far from what I would consider a "standard" LWE instance). The idea is to first use the binary distinguisher approach from prior work (which focuses on identifying the zero vs. non-zero indices of the errors) and then applies a post-processing step that identifies whether two non-zero entries in the candidate secret are equal or not (to identify the set of coordinates that are 1 vs. -1).


**Weaknesses:**

I found the contributions in the paper to be somewhat incremental relative to the previous work. The main improvements seem to be better preprocessing relative to the previous Picante system. While I appreciate that there is a significant speed-up in the new approach for generating LWE samples from a small initial challenge, the core idea is very similar: resample followed by lattice reduction. A more refined application of lattice reduction compared to Picante yields the performance improvement.

The second feature that distinguishes this work from prior work is the extension to ternary and sparse Gaussian secret key distributions. The extension to ternary builds on the binary distinguisher approach of prior work and essentially add a post-processing distinguisher that decides whether two components are equal/unequal. This technique is very tailored to the ternary case, and it is not clear how general it is or whether this has implications on more standard LWE secret key distributions (e.g., uniformly random or (non-sparse) discrete Gaussian).

From a cryptographic perspective, I view LWE with sparse binary/ternary secret keys to be a very risky assumption. The known reductions from LWE to worst-case lattice problems (e.g., gap shortest vector problem) apply only to LWE with uniform secrets (and also LWE with discrete Gaussian secrets). I do not see a solid basis for the hardness of LWE with very sparse binary/ternary secrets, except for the fact that we do not yet know how to break these assumptions. From this perspective, I am not sure if the results in the paper provide insight on the hardness of the standard LWE assumption where we have better evidence for hardness (and which is the basis of schemes in the NIST post-quantum standardization process).

The paper describes attacks on LWE with sparse Gaussian secrets, but this seems like an unusual choice. Normally, when we consider LWE with Gaussian secrets, there is no sparsity condition. Are there concrete settings where this version of the LWE assumption is used? The attack here seems to rely on identifying the non-zero entries (using the binary distinguisher approach from earlier) and then brute force guessing the values in the non-zero indices. This seems rather inefficient and unlikely to scale to even somewhat-dense solutions. The attacks in the paper only consider a very sparse regime, which again corresponds to a setting where I consider an attack to not be particularly surprising.

From the machine learning perspective, I think the paper mostly relies on existing models and architectures as opposed to introducing new architectures tailored for cryptanalyzing LWE. I think this is a fine approach to take, albeit one with less novelty from a machine learning perspective.

**Questions:**

As mentioned above, using a *sparse* Gaussian secret key distribution seems unusual for LWE? Do you have examples where this is used or preferred over a binary/ternary key distribution (sparse Gaussian secrets are not captured by reductions to worst-case lattice problems, and do not seem to confer better efficiency than just using binary/ternary keys).

When you compare the running time of your attack to the estimated security level of the instances you consider, can you also report the running time in terms of the number of bit operations (or CPU cycles) for the attack? From Table 15, it looks like you assume full parallelization, so it is not clear how the running time translates to the actual amount of compute used. Does your algorithm suggest that the security level for small LWE instances (as determined by the LWE estimator) is much lower than currently reported? How big is this gap? I think analyzing this would require computing the actual amount of compute (taking into consideration the parallelism) used in the attack and not just a run time measurement.

When you compare the effectiveness of your attack against known classical algorithms, are you comparing against the best state-of-the-art attacks (it seems the lattice estimator suggest different algorithms in Table 16)? Are the algorithms you compare against tuned for standard LWE instances (e.g., with uniform or Gaussian secret keys)? I would like to understand the extent to which this is an apples-to-apples comparison.

On a similar note, are you comparing against approaches with similar levels of compute? Your algorithm assumes full parallelization; does this apply to the previous algorithms you compare against?

**Limitations:**

I found the comparisons with other classical approaches to be one of the more interesting aspects of this work, and would have liked to see some of this discussion in the main body of the paper. It would be great if the paper could provide some more fine-grained information on the amount of compute needed for the ML-based approaches and what the implications these attacks have for improving our hardness estimates for LWE with sparse secret distributions.

---

> ### Author Rebuttal · Authors · 2023-08-08
>
> **I found the contributions in the paper to be somewhat incremental relative to the previous work.**
>
> 6 main contributions are listed in the introduction, none of which are incremental. Besides the new secret recovery technique for sparse binary, ternary, and small Gaussian secrets, and the improved data preprocessing techniques (40x faster and 20% more effective), Verde also includes:
> * Results for increased dimension n and reduced modulus q that outperform classical attacks, and achieve a new state-of-the-art on medium-to-hard LWE instances with small sparse secrets
> * NoMod, a framework for understanding the success of ML-based attacks
> * Theoretical analysis to explain successful recovery
> * A new encoder-only model architecture, with comparison to old architecture
>
> **The extension of secret recovery] technique is very tailored to the ternary case**
>
> We believe that the new distinguisher can work with any small secret. Suppose secret coordinates can take $k$ different values, in dimension $n$, there are $k^n$ possible secrets. The new distinguisher clusters the $n$ secret positions into $k$ classes having the same secret value. For uniform small secrets, the number of possible secrets becomes $k!$.
>
> If the secret distribution is not uniform, we can leverage the frequency estimates that the distinguisher provides for each class. For instance, for Gaussian or binomial secrets, where $$Prob(0) > Prob(1)=Prob(-1) > Prob(2)=Prob(-2) \dots$$
>
> we can associate the largest class to zero, the two next to $-1$ and $1$, &c. This results in $2^\frac{k-1}{2}$ possible secrets. (Also see the last point from the discussion in our general rebuttal)
>
> **From a cryptographic perspective, I view LWE with sparse binary/ternary secret keys to be a very risky assumption.**
>
> We agree that binary and sparse binary secrets are much weaker than random secrets and that the associated hardness assumption is suspicious.  Yet, they are desirable solutions from a practical point of view, because they run faster, are easier to implement, and allow bootstrapping in homomorphic encryption. For this reason, in the absence of concrete attacks such as the ones our paper suggests, practitioners will push ahead with implementations of cryptosystems based on these risky assumptions.  That is precisely why it is so important to develop attacks like Verde, so the practical vulnerabilities of such systems can be better understood.
>
> **using a sparse Gaussian secret key distribution seems unusual for LWE**
>
> The NIST 2022 standardized schemes use a binomial secret distribution, very close to the Gaussian distribution with width 3, but not sparse.  The 2018 HomomorphicEncryption.org standard proposes both ternary secrets and Gaussian secret distribution with width ~3.  The HE standard does not propose sparse versions, but the community continues to push for sparse versions in the standard, to enable bootstrapping and performance improvements.  We study sparse secrets in our paper because that’s where our current techniques apply best, with the aim to extend to general binary, ternary, and Gaussian secrets in future work.  Sparse ternary secrets are explicitly proposed in several HE application papers, cited in our paper.
>
> **it looks like you assume full parallelization, so it is not clear how the running time translates to the actual amount of compute used.**
>
> In table 15, parallelization only applies to preprocessing, and assumes that each $n\times n$ matrix is reduced on one CPU core. The number of matrix lines needed for VERDE is $2$ million, and so for $n=256$ the number of cores is $2,000,000/256= 7813$. For $n=256$ and $\log q > 16$, our attack only outperforms uSVP because we can trade compute for time during preprocessing and uSVP cannot. For smaller $q$, we could not recover the secret using uSVP. Also, we could recover secrets for $n=512$ and $h=63$., but know of no concrete uSVP attack that achieves such results.
>
> For VERDE, estimating the number of cycles is made difficult by the training stage, which is performed on a GPU.  For uSVP, the LWE estimator gives estimates that can be compared to our preprocessing stage, but they are only estimates, not successful attacks, and it is hard to translate them into concrete running times for comparison. Comparing concrete attack time on the same machines, using the same implementation of BKZ, is the best we can achieve.
>
> **are you comparing against the best state-of-the-art attacks?**
>
> See Appendix A.7 for a discussion of this exact point (lines 620-641, pages 17-18).  We introduce 3 improvements to existing uSVP implementations, to get them to run and allow for apples-to-apples comparison. We provide two benchmarks:
> 1. Comparing to predictions from the LWE Estimator,
> 2. Improving and running implementations of uSVP attacks to validate estimates from the Estimator and provide direct comparisons to our approach in the same hardware/software set-up.  (The estimator does not actually run successful attacks.)
>
> **Your algorithm assumes full parallelization; does this apply to the previous algorithms you compare against?**
>
> PICANTE uses full parallelization of the preprocessing step. SALSA is not comparable, because it operates without preprocessing, and on much smaller instances. For the preprocessing step, table 2 provides a comparison between PICANTE and VERDE preprocessing run on the same machines. For the training step, we are using similar architectures and GPUs as PICANTE (V100 with 32GB video memory). Levels of compute should be in the same ballpark.
>
> **I found the comparisons with other classical approaches to be one of the more interesting aspects of this work, and would have liked to see some of this discussion in the main body of the paper.**
>
> We agree it would be better to include the comparison work in the main body of the paper, and we can provide more fine-grained info and move that into the main body if accepted.

---

### Author Rebuttal · Authors · 2023-08-08

We thank the reviewers for their comments and suggestions. In this general rebuttal, we address two questions that were mentioned in several reviews: how this research impacts the security of LWE, and how to perform full secret recovery of Gaussian secrets.


## 1- General context for VERDE and how it impacts LWE
**(Reviewers wmYQ, XtRr, j97t)**

With respect to problem complexity, instances of cryptographic problems like LWE fall broadly into three buckets:
* Easy: solvable via exhaustive search
* Medium to hard: requiring significant to unrealistic resources to solve
* Standardized: believed secure

SALSA Verde attacks **medium-to-hard** LWE problems (parameterized by dimension $n$, Hamming weight $h$, modulus $q$) and outperforms State-Of-The-Art (SOTA) attacks for LWE with small sparse secrets. Verde is not attacking toy problems (like SALSA did), nor is Verde attacking the NIST standard directly. Rather, Verde provides improved attacks using tools from AI, improving our understanding of the security of proposed LWE-based cryptosystems.

For example, Verde recovers secrets when dimension $n=512$ and Hamming Weight $h=63$ with modulus $q$, $\log q= 41$ (more than $2^{270}$ possible binary secrets; brute force attacks are impossible). In this setting, SOTA attacks (e.g. uSVP) did not succeed for $n=512$, but for smaller dimensions, $n=256$ and $n=350$, we provide extensive comparisons in Appendix A.7 showing Verde’s  speed-up over SOTA.  Thus, Verde is the new SOTA for attacking **medium-to-hard** LWE lattice problems with sparse binary secrets.

We will clarify this in the introduction and add comparison with brute force attacks.


## 2. Full Gaussian secret recovery
**(Reviewers StdV, heZ1)**

Three different techniques can be used for the full recovery of Gaussian secrets.

First, we can use the non-zero secret bit recovery techniques described in the paper to discover the $h$ non-zero coordinates of **s** ($i_1,\dots  i_h$ henceforth). When computing $b=\textbf a\cdot \textbf s+e$, all other coordinates of **a** have no impact (they are multiplied by zero in the dot product). Therefore, we can reduce the problem to an equivalent LWE instance of dimension $h$: replacing **a** by $\textbf a_{red} = (a_{i_1}, \dots a_{i_h})$, **s** by $\textbf s' =( s_{i_1}, \dots s_{i_h})$ and observing that $b = \textbf a_{red}\cdot \textbf s’ +e$. The reduced problem becomes non sparse, and has larger error, but if the dimension $h$ is low (i.e.is the secret is sparse enough) , we can recover **s’** (and therefore **s**) using classical lattice reduction attacks.

Second, we can adapt the ternary attack to the Gaussian case. For small Gaussian secrets, the values of non-zero secret bits follow a Gaussian distribution with a standard deviation of $3$. This means that $99.7$% of non-zero secret bits take values between $-9$ and $9$. For the values of $h$ considered in this paper, we may safely assume that all secret bits are in $[-9,9]$. The two-bit distinguisher, introduced for ternary secrets, allows us to classify all secret positions into $k<<n$ groups of bits, which share a common secret value (i.e. zero "bits", 1 "bits", -1 "bits", 2 "bits" ...). In the sparse case, this allows to recover the $0$ class (which will be the largest group). Then, we can either:
* if $k$ is small, enumerate and test all possible secrets (there are $\frac{18!}{(18-k)!}$ of them)
* assign $-1$ and $1$ to the two largest remaining classes ($2$ possible choices), then $-2$ and $2$ to the next, etc… and test the corresponding $2^{\frac{k-1}{2}}$ secrets
* mix the two previous approaches: guess the bits with smallest absolute value from their frequencies, and enumerate on the smallest ones.

Using these techniques, we build a short list of possible secrets, that can be tested on LWE data using the methods presented in section 3, lines 158 et seq.

Third, we can improve the ternary distinguisher to distinguish secret coordinates of opposite signs. The ternary distinguisher can tell whether two secret values are equal by swapping two positions of **a** ($a_i$ and $a_j$), and comparing the model predictions at $\textbf a(\dots a_i \dots a_j \dots)$ and $\textbf a’(\dots a_j \dots a_i \dots...)$. If the corresponding secret bits, $s_i$ and $s_j$ are equal, the prediction of **a** and **a’** should be equal, or very close. Now consider **a”**$(\dots -a_j \dots  -a_i \dots)$. Model predictions for **a** and **a”** should be close if the corresponding secret values are opposite ($s_i =  - s_j$). Coupled with the ternary distinguisher, this can help classify all the $k-1$ values into $\frac{k-1}{2}$ pairs, and improve recovery.

We will add a discussion of full Gaussian secret recovery in the revised version of the paper.

---

### Decision · Program_Chairs · 2023-09-21

**Decision:**

Accept (poster)

**Comment:**

This paper proposes new machine learning algorithms for solving LWE type problems.
The paper focuses on LWE instances with sparse or small secrets and gives improved techniques for solving those.
As noted by some of the reviewers, it is not quite clear how generalizable the techniques are, or whether the considered LWE problems are really the most relevant in practice.
But ML-based cryptanalysis of number-theoretic problems is a fairly under-explored problem, so I think it is good to see more work towards this direction (even if it currently does not impact the parameter choices for practical schemes).